# A Hierarchical Circuit Symbolic Discovery Framework for Efficient Logic Optimization

**Yinqi Bai[1], Jie Wang[1][†], Xialiang Tong[2], Longdi Pan[1], Jilai Pan[1], Mingxuan Yuan[2], Jianye Hao[3]**

[1]MoE Key Laboratory of Brain-inspired Intelligent Perception and Cognition,
University of Science and Technology of China
[2]Huawei Technologies Co., Ltd.
[3]College of Intelligence and Computing, Tianjin University
byq000324@mail.ustc.edu.cn,

## Abstract

The efficiency of Logic Optimization (LO) has become one of the key bottlenecks in chip design. To prompt efficient LO, many graph-based machine learning (ML) methods, such as graph neural networks (GNNs), have been proposed to predict and prune a large number of ineffective subgraphs of the LO heuristics. However, the high inference cost and limited interpretability of these approaches severely limit their wide application to modern LO tools. To address this challenge, we propose a novel **H**ierarchical **C**ircuit **S**ymbolic Discovery Framework, namely HIS, to learn a *lightweight* and *interpretable* symbolic function that can *accurately* identify ineffective subgraphs for efficient LO. Specifically, HIS proposes a hierarchical tree structure to represent the circuit symbolic function, where every layer of the symbolic tree performs an efficient and interpretable message passing to capture the structural information of the circuit graph. To learn the hierarchical tree, we propose a circuit symbolic generation framework that leverages reinforcement learning to optimize a structure-aware Transformer model for symbolic token generation. To the best of our knowledge, HIS is *the first* approach to discover an efficient, interpretable, and high-performance symbolic function from the circuit graph for efficient LO. Experiments on two widely used circuit benchmarks show that the learned graph symbolic functions outperform previous state-of-the-art approaches in terms of efficiency and optimization performance. Moreover, we integrate HIS with the Mfs2 heuristic, one of the most time-consuming LO heuristics. Results show that HIS significantly enhances both its efficiency and optimization performance on a CPU-based machine, achieving an average run-time improvement of 27.22% and a 6.95% reduction in circuit size. The code is available at https://github.com/MIRALab-USTC/HIS.

## 1 Introduction

The modern chip design workflow leverages a variety of Electronic Design Automation (EDA) tools to efficiently and reliably synthesize, simulate, test, and verify different circuit designs (Huang et al., 2021). Logic Optimization (LO) is one of the most important EDA tools in the front-end workflow (Berndt et al., 2022; Pasandi et al., 2023). Specifically, LO aims to optimize circuits—modeled by directed acyclic graphs—with functionality-equivalent transformations and reduced size and/or depth. It is crucial to well tackle the LO task as it can significantly improve the circuits' Quality of Results (QoR), i.e., various metrics such as size, level, and edge to evaluate the quality of designed chips. (De Abreu et al., 2021; Bertacco et al., 1997). However, the LO task is a challenging $\mathcal{NP}$-hard problem (Micheli, 1994; Farrahi & Sarrafzadeh, 1994), which makes it extremely hard to tackle. To approximately tackle the LO task, many effective LO heuristics such as Mfs2 (Mishchenko et al., 2011), Resub (Brayton, 2006), and Rewrite (Bertacco et al., 1997) have been developed. These

---

[†]Corresponding author. Email: jiewangx@ustc.edu.cn.

heuristics follow a common paradigm in which specific transformations are sequentially applied to the subgraph rooted at each node (i.e., node-level transformations) for all nodes of an input circuit.

The efficiency of LO heuristics in LO tools has become one of the key bottlenecks in chip design, thus significantly impacting the final circuit performance and Time-to-Market, i.e., the overall duration for developing and commercializing new chips (Neto et al., 2021; Sabbavarapu et al., 2014; Reddy et al., 2014). However, previous studies found that executing LO heuristics can be highly time-consuming due to a large number of ineffective and redundant node-level transformations (Wang et al.). To address this, they propose a pruning framework, which leverages a key scoring function to identify and avoid the ineffective transformations. Specifically, (Li et al., 2023) proposes a lightweight mathematical expression as the scoring function. However, this method fails to capture the rich structural information of the subgraph, resulting in limited optimization performance. Recently, (Wang et al.) proposes a well-designed graph neural network (GNN) model, which can effectively capture the subgraphs' structural information to accurately identify ineffective transformations. Nevertheless, the high inference cost and limited interpretability of GNNs significantly restrict their adoption in modern LO tools.

To address these challenges, we propose a novel Hierarchical Circuit Symbolic Discovery Framework, namely HIS, to learn a lightweight and interpretable symbolic function from the circuit subgraph—always modeled by a computation tree—that can accurately identify ineffective transformations for efficient LO. The key technical challenge lies in designing a symbolic function that can effectively capture subgraph structural information. Inspired by the message-passing mechanism in GNNs, HIS proposes a hierarchical symbolic function representation, where each layer performs an interpretable and computationally efficient form of message aggregation to capture the multi-level structural information, as shown in Figure 1. To learn hierarchical symbolic functions, we introduce a circuit symbolic generation framework. In this framework, a structure-aware Transformer is employed to effectively encode tree-structured information and generate a distribution over symbolic subtrees at each layer. Subtrees sampled from this distribution are merged to form hierarchical symbolic trees, which are then evaluated using a group reward. The reward signal is leveraged to optimize the model through a policy gradient algorithm. Ultimately, the target hierarchical symbolic tree is generated according to the optimized model.

Experiments on two widely used benchmarks show that the symbolic scoring functions learned by our HIS outperform previous state-of-the-art approaches in terms of efficiency and optimization performance. Moreover, we incorporate HIS with the Mfs2 heuristic—the most time-consuming one among commonly used LO heuristics. The empirical results on widely used circuit benchmarks demonstrate that HIS achieves an average runtime improvement of 27.22% and a 6.95% reduction in circuit size compared with the default Mfs2 heuristic. Furthermore, our HIS learned hierarchical symbolic functions offer strong interpretability, revealing how specific structural patterns in the circuit graph impact the final node embedding.

We summarize our major contributions as follows: (1) To the best of our knowledge, HIS is *the first* approach to discover an efficient, interpretable, and high-performance graph symbolic function for efficient LO. (2) The major technical contribution of HIS is the novel hierarchical symbolic tree representation that enables interpretable and efficient message aggregation to capture the structural information of circuit graphs. (3) Experiments show that the learned interpretable symbolic functions outperform state-of-the-art approaches in terms of efficiency and optimization performance.

## 2 BACKGROUND

**Logic Optimization (LO)** Driven by Moore's law, the complexity of chip design has increased exponentially (Khailany, 2020; Lopera et al., 2021; Huang et al., 2021; Mirhoseini et al., 2021; Ren & Hu, 2023; Bai et al., 2026). To address this growing complexity, modern design workflows integrate a suite of Electronic Design Automation (EDA) tools to synthesize, simulate, test, and verify different circuit designs efficiently and reliably. Among these tools, logic optimization (LO)—which optimizes circuits represented as Boolean networks—serves as a key component. LO typically involves two stages: pre-mapping optimization and post-mapping optimization (Hosny et al., 2020; Ren & Hu, 2023; Wang et al., 2024; Brayton et al., 2010). In the pre-mapping stage, heuristics such as Rewrite (Bertacco et al., 1997), Resub (Brayton, 2006), and Refactor (Brayton, 1982) are used to optimize the input circuit. The optimized logic circuits are then mapped onto the target technology

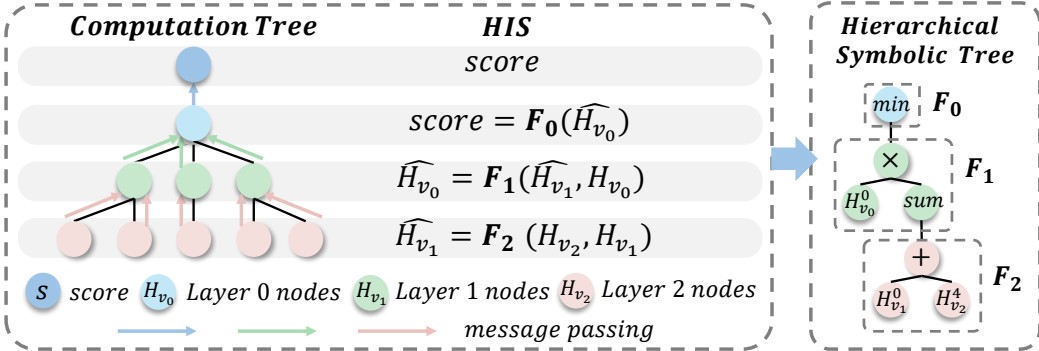

Figure 1: Our HIS framework learns a hierarchical symbolic tree which performs an interpretable and efficient message aggregation motivated by the graph neural networks (GNNs).

library, e.g., a standard-cell netlist (Brayton & Kam) or k-input lookup tables (Mishchenko et al., 2007). Consequently, the post-mapping heuristics like Mfs2 (Mishchenko et al., 2011) are employed to further enhance the mapped circuit.

**The pruning framework for LO Heuristics** Many effective LO heuristics have been developed to tackle the LO task. These heuristics follow the same paradigm as illustrated in Figure 4. Specifically, they apply specific transformations to a subgraph rooted at each node (i.e., the node-level transformations) sequentially for all nodes in an input circuit. These LO heuristics constitute a cornerstone of logic optimization, enabling substantial improvements in circuit quality. However, recent work (Wang et al.) has shown that a large number of node-level transformations in many LO heuristics are ineffective, which makes applying these heuristics highly time-consuming. To address this challenge, several studies (Wang et al.; Li et al., 2023) have proposed a pruning framework, which leverages a scoring function to identify and avoid transformations on those ineffective nodes to improve the efficiency of LO heuristics. Within the pruning framework, the accuracy and efficiency of the scoring function significantly determine the optimization performance and runtime of LO heuristics. Therefore, it is crucial to discover an accurate and efficient scoring function.

**Computation Tree of Graph Neural Networks (GNNs).** GNNs have been developed to solve the LO task. Let $\mathcal{G} = (\mathcal{V}, \mathcal{E})$ denote a circuit graph with node set $\mathcal{V}$ and edge set $\mathcal{E}$. For a target node $v \in \mathcal{V}$, the input to the GNN for LO is defined as a subgraph centered at $v$, which can be equivalently represented by its depth-$L$ computation tree $T_v^L$ as shown in Figure 1. Specifically, we set $T_v^0 = v$, and recursively construct $T_v^L$ for $L > 1$ by expanding $T_v^{L-1}$ with the neighbors of all leaf nodes in $T_v^{L-1}$. The GNN encoder takes $T_v^L$ as input and performs iterative message passing to learn node embeddings. Formally, the $L$-th layer of a GNN encoder can be written as:

$$\mathbf{h}_v^{(L)} = UPDATE^{(L)}\Big(\mathbf{h}_v^{(L-1)}, AGGREGATE^{(L)}\big(\mathbf{h}_u^{(L-1)} : u \in \mathcal{N}(v)\big)\Big),$$

where $\mathbf{h}_v^{(L)}$ denotes the embedding of node $v$ at layer $L$, $\mathbf{h}_v^{(0)}$ is initialized from its feature vector, and $\mathcal{N}(v)$ denotes the set of neighbors of $v$.

## 3 RELATED WORK

**Scoring functions for LO heuristics.** A variety of approaches have been developed to address the LO task, which can be broadly categorized into heuristic and machine learning-based methods. Heuristic methods, such as (Li et al., 2023), manually design lightweight scoring functions derived from circuit structure. However, these approaches often fail to capture the rich structural information of subgraphs, resulting in limited optimization performance. In contrast, machine learning methods, such as (Bai et al.) and (Wang et al.), employ graph convolutional networks either to generate scoring functions or to directly serve as them. Nevertheless, their high inference cost and limited interpretability significantly hinder their adoption in modern LO tools. These limitations highlight the need for a scoring function that is accurate, interpretable, and computationally efficient.

**Graph Symbolic Distillation from GNNs** Motivated by the high expressive power but opaque nature of GNNs, prior research has focused on distilling interpretable symbolic functions to approximate their mapping mechanisms. For example, (Cranmer et al., 2020) proposed a framework

that extracts symbolic functions from trained GNNs for scientific discovery. The approach first trains neural network models and then employs a symbolic learning model to approximate both the message-passing and aggregation functions with symbolic representations. However, this method requires extensive process labels to train the symbolic model, which limits its scalability and practicality. More recently, (Kuang et al.) proposed an end-to-end framework to learn interpretable symbolic policies from a general bipartite graph representation. Nevertheless, we observe that this method struggles to effectively capture circuit structural information, often leading to suboptimal performance. To address these limitations, we propose a hierarchical symbolic tree representation that learns structural symbolic functions in an end-to-end fashion without relying on process labels.

## 4 METHOD

In this section, we provide a comprehensive description of our Hierarchical Circuit Symbolic Discovery (HIS) framework. We begin by introducing the hierarchical circuit symbolic tree representation in Section 4.1, where the structural properties of circuits are captured in a symbolic form. Then, we present the circuit symbolic generation framework (see Figure 2), which proposes a reinforcement learning based approach for discovering the symbolic tree, as detailed in Section 4.2.

### 4.1 THE HIERARCHICAL CIRCUIT SYMBOLIC TREE REPRESENTATION

**Motivation** To prompt efficient LO, graph neural networks (GNNs) (Wang et al.) have been applied as the scoring function to predict and prune the ineffective node-level transformations. While this method achieves high performance on node classification, the complex architecture and extensive parameters significantly restrict its deployment in pure CPU-based industrial scenarios. To address this limitation, we propose to distill the GNN into a symbolic representation for efficient deployment. However, it is challenging to learn a lightweight symbolic function that preserves expressive graph representation capabilities. Motivated by the layered message-passing mechanism of GNNs, which integrates structural information from neighboring nodes, we propose the hierarchical circuit symbolic function representation. This representation comprises multi-layer functions that emulate the message-passing process across layers, enabling the resulting symbolic functions to capture both local and global graph structural information effectively for node classification.

**Graph features and Symbolic library** Given a circuit subgraph rooted at node $v_0$, we follow (Wang et al.) and represent it as an $L$-layer computation tree $T_{v_0}^L$ (see more details about the subgraph construction process in Appendix D.4). By traversing all nodes in the circuit graph, we can thus construct a training dataset $D = \{(T_{v_i}^L, y_i)\}_{i=1}^n$. In our experiments, we set $L = 2$, which is consistent with the 2-layer GNN configuration adopted in (Wang et al.). To learn a symbolic function from the graph, we first define the graph features and symbolic library. Each node $v \in T_{v_0}^L$ is initialized with a 5-dimensional structural feature vector $\mathbf{h}_v$, as defined in (Bai et al.). Let $\mathbf{v}_i = \{v_i^j\}_{j=1}^{n_i}$ denote the set of nodes at layer $i$. Then the node features at layer $i$ can be represented as

$$\mathbf{H}_{\mathbf{v}_i} = \begin{bmatrix} \mathbf{h}_{v_i^1} & \cdots & \mathbf{h}_{v_i^{n_i}} \end{bmatrix}^\top \in \mathbb{R}^{n_i \times 5}.$$

Finally, the graph features are represented as the union of node features across all layers

$$\mathcal{F} = \bigcup_{i=0}^{L} \{\mathbf{H}_{\mathbf{v}_i}^0, \cdots, \mathbf{H}_{\mathbf{v}_i}^5\},$$

where $\mathbf{H}_{\mathbf{v}_i}^j \in \mathbb{R}^{n_i}$ denotes the $j$-th column of the node feature matrix at layer $i$. Moreover, considering the symbolic library, we employ $\{+, -, \times, \div, \log, \exp\}$ as the mathematical operators and $\{0.1, 0.2, 0.5\}$ as constants. To aggregate neighborhood information, we follow (Kuang et al.) and employ four unary operators, $\{\min, \max, \text{mean}, \text{sum}\}$, each mapping features from layer $i$ to layer $i-1$, i.e., $\mathbb{R}^{n_i} \to \mathbb{R}^{n_{i-1}}$. The aggregation is performed according to edge connectivity, ensuring that only features of adjacent nodes are combined. These aggregation operators play the same role as those in GNNs, which can effectively capture the graph structural information. Finally, the symbolic library comprises mathematical operators, aggregation operators, and constants.

**Hierarchical Symbolic Tree** To represent the hierarchical symbolic functions, we employ a tree-structured representation in which the leaf nodes correspond to features or constants, and the internal

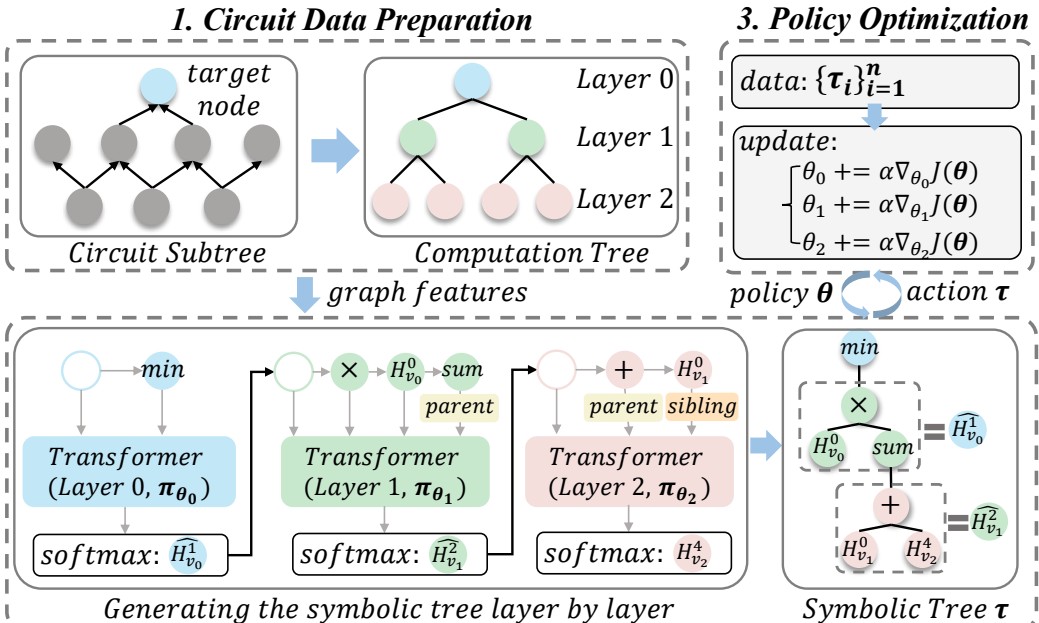

Figure 2: Illustration of the Circuit Symbolic Generation framework. The circuit is first represented as a computation tree to provide graph features. Then, a hierarchical symbolic function is generated layer by layer using Transformer-based policies that incorporate both parent and sibling information. Finally, the model parameters are optimized via policy gradient methods.

nodes denote mathematical operators (Kuang et al., 2024; Sun et al., 2023; Petersen et al., 2020). Unlike traditional symbolic functions that take node features as input, our circuit symbolic tree directly operates on the circuit computation tree to aggregate the structural information for node classification. Specifically, given an $L$-layer computation tree $T_{v_0}^L$ rooted at node $v_0$, the hierarchical symbolic tree is organized into $L$ layers (see Figure 1). For each node at layer $i$ of the computation tree, the learned function $\mathbf{F}_i$ aggregates messages from its neighboring nodes to update its feature representation. In general, the aggregation function in layer $i$ can be expressed as

$$\begin{cases} \text{score} = \mathbf{F}_i\big(\hat{\mathbf{H}}_{\mathbf{v}_i}\big), & \text{if } i = 0, \\ \hat{\mathbf{H}}_{\mathbf{v}_{i-1}} = \mathbf{F}_i\big(\hat{\mathbf{H}}_{\mathbf{v}_i}, \mathbf{H}_{\mathbf{v}_{i-1}}\big), & \text{if } i > 0, \end{cases}$$

where $\hat{\mathbf{H}}_{\mathbf{v}_{i-1}} \in \mathbb{R}^{n_{i-1} \times d}$ denotes the updated node feature matrix at layer $i-1$, and $\mathbf{F}_i$ represents $d$ aggregation functions that update the features of layer $i$ nodes based on their current features and the features of nodes in layer $i-1$. The parameter $d$, which denotes the dimensionality of the updated feature vectors, is set to 10 in our experiments. By combining the learned symbolic functions $\mathbf{F}_i$, we can construct a hierarchical symbolic tree. This symbolic tree captures the structural information of the circuit through layer-wise aggregation while retaining the efficiency of symbolic operations, which is deployable in industrial settings.

### 4.2 The Circuit Symbolic Generation Framework

**Symbolic Sequence Formulation** In this section, we provide a detailed explanation of the learning process for symbolic trees at each hierarchical layer. Previous works usually employ a pre-order traversal sequence $\tau = \{\tau_1, \tau_2 \cdots \tau_n\}$ to represent the symbolic tree. Therefore, the task of generating symbolic trees can be formulated as a sequence generation task. At each generation step, we output a categorical distribution over all possible tokens to sample the current token $\tau_i$. Finally, we can generate a symbolic sequence $\tau$ with the probability given by

$$p_{\boldsymbol{\theta}}(\tau) = \prod_{i=1}^{|\tau|} p_{\boldsymbol{\theta}}(\tau_i | \tau_1, \cdots, \tau_{i-1}),$$

where $p_{\boldsymbol{\theta}}(\tau)$ is the probability of generating the sequence $\tau$, $\boldsymbol{\theta}$ is the generation model parameter, and $p_{\boldsymbol{\theta}}(\tau_i | \tau_1, \cdots, \tau_{i-1})$ is the conditional probability of generating token $\tau_i$ at $i$-th step.

**Transformer Model for Symbolic Generation** To generate the hierarchical symbolic tree, we design $L$ encoder-only Transformer models, each model $\pi_{\theta_i}$ responsible for generating the symbolic functions at layer $i$. At each generation step $k$, the layer $i$ generation model takes the previously sampled token sequence as input and predicts the next token $\tau_k^i$. However, a standard Transformer is limited in capturing the tree-structural dependencies of symbolic expressions, which often leads to suboptimal results (Holt et al., 2023; Petersen et al., 2020). To overcome this limitation, we design a tree-aware embedding aggregation mechanism inspired by (Kuang et al.). Specifically, during the generation period of token $\tau_k^i$, we first identify its parent and sibling tokens, denoted as $\tau_{p_k}^i$ and $\tau_{s_k}^i$, and then encode them as $\text{Parent}(\tau_{p_k}^i) = \beta_p$ and $\text{Sibling}(\tau_{s_k}^i) = \beta_s$, respectively. After passing through several encoder layers—each consisting of a multi-head attention and a feed-forward network—we compute the representation of $\tau_k^i$ by averaging the embedding vectors of $\tau_{p_k}^i$ and $\tau_{s_k}^i$. We then apply a softmax function to this representation to obtain the probability distribution $p_{\theta_i}(\tau_k^i | \tau_1^i, \ldots, \tau_{k-1}^i)$, which is used to generate the token $\tau_k^i$ at step $k$.

**Training Model via Reinforcement Learning** Since the symbolic tree is not differentiable with respect to the model parameters $\boldsymbol{\theta} = (\theta_0, \cdots, \theta_L)$, we formulate the sequence generation as a reinforcement learning problem. Specifically, we formulate the transformer models as the policy network, treat the sampled tokens as states, and consider each generated token as an action. Furthermore, we regard a complete sequence of tokens as an episode, and define the reward as a terminal signal that is computed only upon the completion of the expression. In each episode, we sample a group of $m$ symbolic expressions and use them to optimize the policy parameters via Proximal Policy Optimization (PPO). Specifically, we define the objective function as

$$J(\boldsymbol{\theta}) = \mathbb{E}_{\boldsymbol{\tau} \sim p(\boldsymbol{\tau}|\boldsymbol{\theta})} \left[ \min\left( \frac{p_{\boldsymbol{\theta}}(\boldsymbol{\tau})}{p_{\boldsymbol{\theta}_{\text{old}}}(\boldsymbol{\tau})} A_{\boldsymbol{\theta}_{\text{old}}}(\boldsymbol{\tau}), \ \text{clip}\left( \frac{p_{\boldsymbol{\theta}}(\boldsymbol{\tau})}{p_{\boldsymbol{\theta}_{\text{old}}}(\boldsymbol{\tau})}, 1 - \epsilon, 1 + \epsilon \right) A_{\boldsymbol{\theta}_{\text{old}}}(\boldsymbol{\tau}) \right) \right],$$

where $\boldsymbol{\tau} = (\tau^1, \tau^2, \ldots, \tau^L)$ denotes the hierarchical symbolic tree obtained by sequentially merging the generated symbolic trees $\tau^i$ from each layer, $p_{\boldsymbol{\theta}}(\boldsymbol{\tau}) = \sum_{i=0}^{L} p_{\theta_i}(\tau^i)$ denotes the probability of generating the hierarchical symbolic tree $\boldsymbol{\tau}$ under the policy parameters $\boldsymbol{\theta}$, $A_{\boldsymbol{\theta}_{\text{old}}}(\boldsymbol{\tau})$ is the advantage function, and $\epsilon$ is the clipping threshold that constrains the policy update. Unlike traditional PPO, which requires training a resource-intensive critic network for advantage prediction, we compute the advantage as the sequence reward relative to the group mean reward, achieving both lower resource consumption and more stable training. The advantage is defined as

$$A_{\boldsymbol{\theta}}(\boldsymbol{\tau}) = \frac{r(\boldsymbol{\tau}) - \bar{r}}{\sigma_r},$$

where $\bar{r} = \mathbb{E}_{\boldsymbol{\tau} \sim p(\boldsymbol{\tau}|\boldsymbol{\theta})}[r(\boldsymbol{\tau})]$ and $\sigma_r = \sqrt{\mathbb{E}_{\boldsymbol{\tau} \sim p(\boldsymbol{\tau}|\boldsymbol{\theta})} \left[ (r(\boldsymbol{\tau}) - \bar{r})^2 \right]}$ denote the mean and standard deviation of rewards in the generated function groups. Given a symbolic tree $\boldsymbol{\tau}$ and the training data $D = \{(T_{v_i}^L, y_i)\}_{i=1}^n$, the reward is computed using the focal loss (Lin et al., 2017), defined as

$$r(\boldsymbol{\tau}) = -\frac{1}{n} \sum_{i=1}^{n} \left[ \alpha y_i (1 - \hat{y}_i)^\gamma \log(\hat{y}_i) + (1 - \alpha)(1 - y_i)\hat{y}_i^\gamma \log(1 - \hat{y}_i) \right],$$

where $\hat{y}_i = \boldsymbol{\tau}(T_{v_i}^L)$ is the predicted score, $\alpha$ is a balancing factor. The detailed training algorithm is illustrated in Algorithm 1. Overall, this reinforcement learning framework allows our model to efficiently discover high-quality symbolic expressions.

# 5 EXPERIMENT

In this section, we conduct extensive experiments to evaluate HIS, which consist of four main parts: **Experiment 1.** Demonstrate the superior performance of our HIS in terms of node classification accuracy and heuristics efficiency. **Experiment 2.** Demonstrate that our method can not only enhance the efficiency but also improve the Quality of Results (QoR) of one of the most time-consuming LO heuristics, Mfs2. **Experiment 3.** Perform ablation experiments to provide further insight into HIS. **Experiment 4.** Show the appealing features of HIS in inference efficiency and interpretability.

**Benchmarks** We evaluate HIS on two widely used public benchmarks, EPFL (Amarú et al., 2015) and IWLS (Albrecht, 2005). The EPFL benchmark comprises 20 circuits, including large-scale cases with up to 214,335 nodes. The IWLS benchmark contains 21 circuits, including challenging cases with up to 1,130 inputs and 1,416 outputs. We defer more benchmark details to Appendix C.1

Table 1: The results show that HIS consistently outperforms all graph-based and node-based baselines in terms of generalization top 50% prediction recall.

| Circuits | Hyp | Square | Multiplier | DesPerf | Ethernet | Conmax |
|---|---|---|---|---|---|---|
| Method | Recall↑ | Recall↑ | Recall↑ | Recall↑ | Recall↑ | Recall↑ |
| COG | **0.87** | 0.81 | 0.82 | 0.81 | 0.55 | 0.75 |
| CMO | 0.79 | 0.94 | 0.87 | 0.79 | 0.59 | 0.73 |
| Effisyn | 0.18 | 0.04 | 0.13 | 0.28 | 0.88 | 0.05 |
| Random | 0.50 | 0.48 | 0.44 | 0.50 | 0.47 | 0.50 |
| HIS (Ours) | 0.82 | **0.94** | **0.94** | **0.83** | **0.99** | **0.75** |

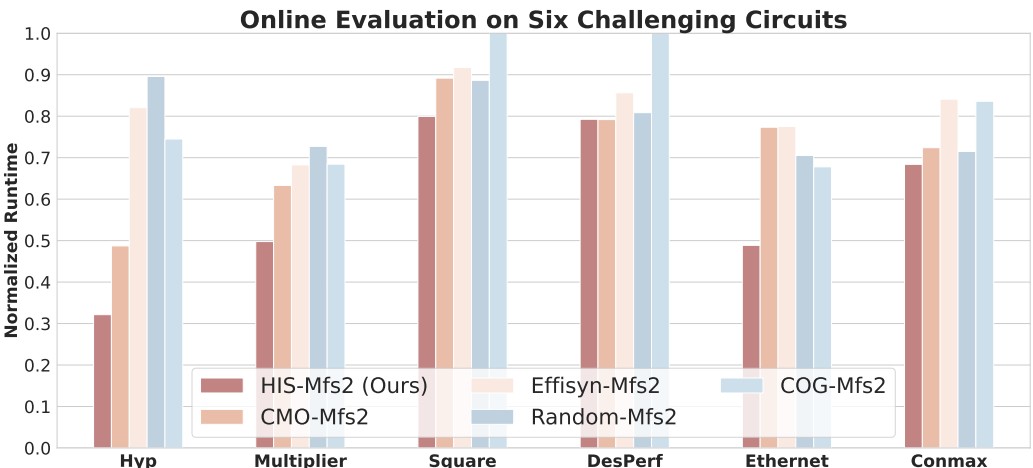

Figure 3: We compare our HIS with four competitive baselines on online runtime. The results demonstrate that our approach achieves significant runtime improvement with the baselines.

**Experimental setup** Throughout all experiments, we use ABC (Brayton et al., 2010) as the backend LO framework. ABC is a state-of-the-art open-source LO framework and is widely used in research on machine learning for LO. Moreover, we choose the Mfs2 (Mishchenko et al., 2011)—one of the most time-consuming LO heuristics—as the backend heuristic. Experiments are performed on a single machine that contains 32 Intel XeonR E5-2667 v4 CPUs, which closely resembles a real-world industrial deployment environment. More details are provided in Appendix D.1.

**Evaluation Metrics and Evaluated Methods** Throughout all experiments, we evaluate our method in two separate phases, i.e., the offline and online phases. **In the offline phase**, we focus on evaluating the prediction recall of effective nodes. The evaluation setup is detailed in two parts as follows: (1) **Evaluation metrics** Under the pruning framework (see Figure 4), we formulate the prediction task as a node scoring problem, where nodes with the top $k$ scores are predicted to be positive. Based on this formulation, we define the **top $k$ accuracy** metric, which measures the fraction of true positive nodes among those predicted as positive, i.e., prediction recall. As shown in Appendix B.1, a higher prediction recall consistently leads to improved QoR. Therefore, achieving high recall is essential for obtaining QoR comparable to the default heuristics. Further details of this metric are provided in Appendix D.1.2. (2) **Evaluated methods** We compare five main approaches: COG (Wang et al.), CMO (Bai et al.), Effisyn (Li et al., 2023), Random and our proposed HIS. COG is a carefully designed 2-layer GNN. CMO is a graph-enhanced symbolic learning method. Effisyn is a human-designed nonlinear function with parameters derived from circuit features. Random is a heuristic method which randomly predicts the score between 0 and 1. Implementation details of

Table 2: We compare the Default Mfs2 heuristic with our HIS-Mfs2 heuristic with the hyperparameter $k$ set as $30\%$, $40\%$ and $50\%$ on six challenging circuits. Optimized Nd denotes the node number (size) of circuits, and Lev denotes the level (depth) of circuits. We define an Improvement metric by $\frac{M(\text{Default}) - M(\text{Ours})}{M(\text{Default})}$, where $M(\cdot)$ denotes the Optimized Nd, Lev, or Time.

| | Hyp | | | | Square | | | |
|---|---|---|---|---|---|---|---|---|
| Method | Lev ↓ | Improvement ↑ (Lev, %) | Time (s) ↓ | Improvement ↑ (Time, %) | Optimized Nd ↓ | Improvement ↑ (Optimized Nd, %) | Time (s) ↓ | Improvement ↑ (Time, %) |
| Default(Mfs2) | 8259.00 | NA | 265.93 | NA | 5701.00 | NA | 21.48 | NA |
| HIS-Mfs2 (0.5, Ours) | 8259.00 | 0.00 | 85.46 | 67.86 | 5703.00 | -0.04 | 10.69 | 50.24 |
| 2HIS-Mfs2 (0.3, Ours) | 5762.00 | 30.23 | 147.62 | 44.49 | 5553.00 | 2.60 | 16.21 | 24.54 |
| 2HIS-Mfs2 (0.4, Ours) | 5762.00 | 30.23 | 192.51 | 27.61 | 5542.00 | 2.79 | 19.97 | 7.03 |

| | Multiplier | | | | DesPerf | | | |
|---|---|---|---|---|---|---|---|---|
| Method | Optimized Nd ↓ | Improvement ↑ (Optimized Nd %) | Time (s) ↓ | Improvement ↑ (Time, %) | Optimized Nd ↓ | Improvement ↑ (Optimized Nd, %) | Time (s) ↓ | Improvement ↑ (Time, %) |
| Default(Mfs2) | 7799.00 | NA | 16.91 | NA | 30853.00 | NA | 28.82 | NA |
| HIS-Mfs2 (0.5, Ours) | 7799.00 | 0.00 | 13.52 | 20.03 | 31035.00 | -0.59 | 22.84 | 20.73 |
| 2HIS-Mfs2 (0.3, Ours) | 7661.00 | 1.77 | 16.48 | 2.50 | 30104.00 | 2.43 | 24.88 | 13.65 |
| 2HIS-Mfs2 (0.4, Ours) | 7659.00 | 1.80 | 20.60 | -21.83 | 29493.00 | 4.41 | 31.71 | -10.03 |

| | Ethernet | | | | Conmax | | | |
|---|---|---|---|---|---|---|---|---|
| Method | Optimized Nd ↓ | Improvement ↑ (Optimized Nd %) | Time (s) ↓ | Improvement ↑ (Time, %) | Optimized Nd, ↓ | Improvement ↑ (Optimized Nd, %) | Time (s) ↓ | Improvement ↑ (Time, %) |
| Default(Mfs2) | 13638.00 | NA | 27.00 | NA | 16509.00 | NA | 19.93 | NA |
| HIS-Mfs2 (0.5, Ours) | 13639.00 | -0.01 | 13.19 | 51.16 | 16760.00 | -1.52 | 13.63 | 31.59 |
| 2HIS-Mfs2 (0.3, Ours) | 13511.00 | 0.93 | 14.84 | 45.06 | 15890.00 | 3.75 | 13.33 | 33.09 |
| 2HIS-Mfs2 (0.4, Ours) | 13509.00 | 0.95 | 19.36 | 28.31 | 15782.00 | 4.40 | 16.77 | 15.87 |

these baselines are deferred to Appendix D.2. **In the Online phase,** we evaluate both the efficiency and QoR of HIS. The evaluation setup is as follows: (1) **Evaluation metrics** For efficiency, we measure the runtime of the heuristics. For QoR, we primarily consider the number of optimized circuit nodes, as this directly influences the final chip area. In addition, we evaluate the circuit depth (i.e., level) of the optimized circuits, which serves as a proxy metric for chip delay. (2) **Evaluated methods** We introduce a new heuristic, X-Mfs2, which incorporates a learned scoring function "X" into the default Mfs2 heuristic. In our experiments, "X" corresponds to the baselines and our HIS.

**Generalization Evaluation Strategy** In practical industrial settings, it is desirable for the trained model to generalize effectively to unseen circuits. To evaluate this capability, we design two generalization strategies. In the first setting, we use the IWLS circuits as the training dataset and select three hard-to-optimize circuits—Hyp, Multiplier, and Square—from the EPFL benchmark as test cases. In the second setting, we reverse the roles by training on the EPFL circuits and testing on three challenging circuits—DesPerf, Ethernet, and Conmax—from the IWLS benchmark. Due to limited space, we provide more details about our designed generalization strategy in Appendix C.2

**Experiment 1. Comparative Evaluation** In this subsection, we compare the offline and online metrics of our HIS framework with the baselines. Following the established generalization strategy, we conduct experiments on six challenging circuits from two widely used open-source benchmarks. **In the offline phase**, we adopt the top 50% accuracy as the evaluation metric. Results in Table 1 show that HIS consistently outperforms all baselines in terms of the prediction recalls, highlighting the superior generalization ability of our method. Moreover, HIS achieves a prediction recall exceeding 80% on the majority of test circuits, indicating that it can preserve most of the effective transformations. **In the online phase**, we primarily focus on evaluating the efficiency of the X-Mfs2 heuristics. To ensure a fair comparison for efficiency, we compare the runtime of different methods when they achieve similar optimization performance. As larger top $k$ accuracy improves the final performance, we employ top 50% for our HIS, top 60% for COG and CMO, and higher $k$ for the Effisyn and Random baselines to achieve comparable optimization performance (see Table 7 in Appendix B.2 for the optimization results). Specifically, the results in Figure 3 indicate that our HIS-Mfs2 achieves an average improvement of 11.96%, 21.82%, 19.24%, and 22.91% over CMO-Mfs2, Effisyn-Mfs2, Random-Mfs2, and COG-Mfs2 in terms of the runtime, respectively. Overall, the offline and online results show that our HIS can not only accurately predict the effective transformations but also outperform all competitive baselines in terms of heuristic efficiency.

**Experiment 2. Improving Efficiency and QoR of the LO heuristic** In this subsection, we conduct experiments on six challenging circuits to demonstrate that our method not only reduces runtime but also improves QoR, measured by the size and level of the optimized circuits. These two metrics are critical in chip design, as they serve as proxies for the final chip area and delay. We first show that

Table 3: The ablation results demonstrate that each component contributes significantly to the overall performance of HIS. Removing any individual module leads to noticeable performance degradation, indicating that the effectiveness relies on the complementary design.

| Circuits | Hyp | Multiplier | Square | DesPerf | Ethernet | Conmax |
|---|---|---|---|---|---|---|
| Method | Recall↑ | Recall↑ | Recall↑ | Recall↑ | Recall↑ | Recall↑ |
| HIS (Ours) | 0.82 | **0.94** | **0.94** | **0.83** | **0.99** | **0.75** |
| w/o hierarchical | 0.81 | 0.91 | 0.74 | 0.77 | 0.81 | 0.72 |
| w/o group optimization | **0.88** | 0.89 | 0.90 | 0.51 | 0.87 | 0.74 |
| w/o tree-structured aggregation | 0.81 | 0.51 | 0.94 | 0.79 | 0.91 | 0.75 |

HIS can enhance the efficiency of the Mfs2 heuristic while maintaining comparable optimization performance. Results in Table 2 demonstrate that our HIS with top 50% accuracy achieves an average runtime reduction of 40.27% with only a marginal 0.38% degradation in circuit size and level across the six test circuits. In particular, HIS attains up to $3.1\times$ faster runtime on the Hyp circuit. Building on this efficiency advantage, we explore applying HIS-Mfs2 sequentially rather than once (denoted as 2HIS-Mfs2) to further improve QoR. Since HIS-Mfs2 runs significantly faster than the default Mfs2 heuristic, this repeated application is computationally feasible. To accelerate runtime, we additionally consider smaller hyperparameter settings, using $k = 30\%$ and $40\%$ instead of $50\%$. When prioritizing optimization performance in real-world scenarios, we can set $k = 40\%$. Table 2 shows that 2HIS-Mfs2 with $k = 40\%$ achieves an average reduction of 7.43% in size and depth while also reducing runtime by 7.82% on the test circuits. When prioritizing runtime in real-world scenarios, results in Table 2 show that 2HIS-Mfs2 with $k = 30\%$ achieves an average reduction of **6.95%** in size and depth, along with a runtime reduction of **27.22%**. Overall, these results demonstrate that the proposed HIS-Mfs2 framework can simultaneously deliver faster runtime and improved QoR, highlighting its potential to yield significant economic benefits in practical chip design.

**Experiment 3. Ablation Study** In this subsection, we perform an ablation experiment to evaluate the individual contribution of each component in HIS. Specifically, the results in Table 3 indicate three key findings as follows. (1) The *'w/o hierarchical'* variant, which learns a complete symbolic tree from the training data end to end rather than layer by layer, led to significant degradation in the prediction recall. This highlights that our proposed hierarchical symbolic function representation can effectively capture circuit structural information for node classification; (2) The *'w/o group optimization'* variant, which employs the single symbolic function's reward rather than the group rewards as the advantage, exhibits a noticeable reduction in prediction recall across the majority of circuits. This demonstrates the effectiveness of the group advantage in improving symbolic expressiveness; (3) The *'w/o tree-structured aggregation'* variant, which omits the aggregation of parent and sibling embeddings during token generation, exhibits a substantial decline in prediction recall. This result highlights the necessity and effectiveness of our proposed approach in capturing tree-structured information for high-performance symbolic trees generation.

**Experiment 4. Strengths for Deployment** In this subsection, we conduct extensive experiments to demonstrate the appealing features of our HIS on inference efficiency and interpretability. Specifically, we present a detailed analysis as follows.

**Inference Efficiency** We compare the inference time of our HIS against both the SOTA graph-based method COG and several lightweight baselines under a pure CPU industrial environment. As shown in Table 5 in Appendix B.3, HIS achieves an average inference speedup of $296\times$ over COG on the EPFL circuits and $254\times$ on the IWLS circuits, while maintaining inference times comparable to other lightweight methods. These results demonstrate that HIS successfully learns a lightweight graph symbolic scoring function that achieves both high prediction recall (see Table 1) and efficient inference, making it well-suitable for deployment in real industrial scenarios.

**Interpretability** We visualize the discovered hierarchical symbolic functions for EPFL and IWLS benchmarks in Table 6 and Figure 5 in the Appendix. Owing to the high interpretability of the symbolic functions, these learned symbolic policies allow researchers to gain deeper insight into the patterns extracted from circuit graphs and to trace how information is aggregated. Specifically, we observe that: (1) All the discovered expressions aggregate information from both the root node and the candidate node, which is consistent with the design in previous GNN-based approaches. (2) The hierarchical symbolic tree structure successfully performs an efficient message-passing across layers, wherein the node features in Layer $i - 1$ are updated through aggregation from Layer $i$.

## 6 Conclusion

To enable efficient Logic Optimization (LO), previous machine learning methods propose to use scoring functions to predict and prune ineffective nodes in LO heuristics. However, the high inference cost and limited interpretability of these approaches severely limit their wide application to modern LO tools. To address this, we propose HIS, a novel Hierarchical Circuit Symbolic discovery Framework that learns efficient, interpretable, and high-performance symbolic functions from the circuit graph. Extensive experiments on two widely used benchmarks show that the learned graph symbolic functions outperform previous state-of-the-art approaches in terms of efficiency and optimization performance. Moreover, HIS significantly enhances both the Mfs2 heuristic's efficiency and optimization performance on a CPU-based machine, achieving an average runtime improvement of 27.22% and a 6.95% reduction in circuit size.

## Ethics Statement

This research does not involve any personally identifiable information. All datasets used are publicly available and widely adopted in the community, and we have verified that their licenses permit research use. In accordance with the ICLR Code of Ethics (`https://iclr.cc/public/CodeOfEthics`), we ensure that our work adheres to principles of fairness, transparency, and responsible AI research. We also disclose that LLMs were used for text polishing, while all conceptual contributions and validation remain the responsibility of the authors in Appendix E.

## Reproducibility Statement

We will provide open access to all source code, configuration files, and preprocessing scripts, together with detailed instructions to reproduce the main experimental results. All datasets employed are publicly available, and we specify the exact versions and preprocessing steps. We report all hyperparameters, model versions, and API parameters in full, and we describe the computational environment (hardware type, GPU model, and software dependencies) in the supplemental material. We also include ablation studies and negative results to ensure transparency. Collectively, these resources and specifications enable reliable and faithful reproduction of our results.

## Acknowledgments

This work was supported in part by National Key R&D Program of China under contract 2022ZD0119801, National Nature Science Foundations of China grants U23A20388 and 62021001. This work was supported in part by Huawei as well. We would like to thank all the anonymous reviewers for their insightful comments.

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

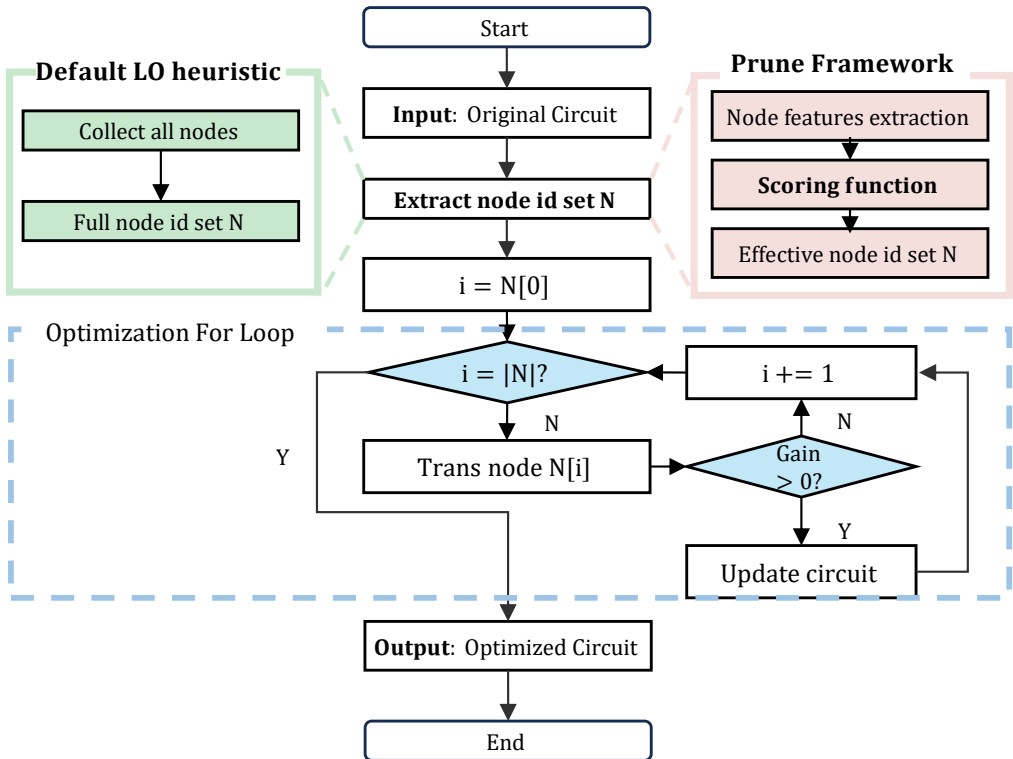

Figure 4: An illustration of the pruning framework for LO heuristics. The scoring function aims to predict and prune the ineffective node-level transformations to prompt efficient LO.

Yilong Xu, Yang Liu, and Hao Sun. Reinforcement symbolic regression machine. In *The Twelfth International Conference on Learning Representations*, 2024. URL https://openreview.net/forum?id=PJVUWpPnZC.

## A   MORE DETAILS OF THE BACKGROUND AND RELATED WORK

**Logic Optimization heuristics** To tackle the LO task, many researchers have developed a rich set of LO heuristics. For instance, researchers have developed Rewrite (Bertacco et al., 1997) and Resub (Brayton, 2006) for pre-mapping optimization, while Mfs2 (Mishchenko et al., 2011) is designed for post-mapping optimization. These LO heuristics follow the paradigm as shown in Figure 4. Specifically, these heuristics traverse the Boolean network in a topological order from PIs to POs and apply transformations to subgraphs rooted at each node sequentially for all nodes. However, previous literature found that these heuristics can be highly time-consuming due to a large number of ineffective transformations. To address this problem, we follow the new heuristics paradigm proposed by (Wang et al.) that can significantly improve the efficiency of LO heuristics by learning a classifier to predict nodes with ineffective transformations and avoid applying transformations on these nodes.

**Circuit Representation** In the LO stage, a circuit is usually modeled by a directed acyclic graph (DAG), where nodes correspond to Boolean functions and directed edges correspond to wires connecting these functions. A Boolean function takes the form $f : \mathbf{B}^n \to \mathbf{B}$, where $\mathbf{B} = \{0, 1\}$ denotes the Boolean domain. Given a node, its fanins are nodes connected by incoming edges of this node, and its fanouts are nodes connected by outgoing edges of this node. The primary inputs (PIs) are nodes with no fanin, and the primary outputs (POs) are nodes with no fanout. The *size* of a circuit denotes the number of nodes in the DAG. The depth (level) of a circuit denotes the maximal length of a path from a PI to a PO in the DAG. The size and depth of a circuit are proxy metrics for the area and delay of the circuit, respectively.

**Deep Symbolic Discovery** Several recent approaches utilize deep learning for symbolic discovery. These methods generally fall into two categories: pre-trained and search-based. The pre-trained symbolic regression methods have shown advantages in fast inference and have successfully discovered large input (with up to twelve) symbolic functions (d'Ascoli et al., 2023; Biggio et al., 2021; Kamienny et al., 2022). However, these methods are limited by high training costs and data generalization challenges. The search-based approach explores the discrete symbolic operator space to identify functions that maximize the fitness with respect to the given dataset. Mainstream symbolic regression frameworks based on this paradigm typically employ sequence prediction models, such as Transformers (Kuang et al.; Holt et al., 2023) and RNNs (Petersen et al., 2020), or leverage Monte Carlo tree search (Sun et al., 2023; Xu et al., 2024). These methods have achieved state-of-the-art performance across multiple benchmarks.

## B  ADDITIONAL RESULTS

### B.1  THE IMPORTANCE OF THE PREDICTION RECALL ON OPTIMIZATION PERFORMANCE

In this subsection, we explore how prediction recalls of effective nodes influence the optimization performance of heuristics. To do this, we assess the performance of the Random method with different values of the hyperparameter $k$, which denotes the percent of nodes to apply transformations. Note that Random is a baseline that randomly assigns a score between $[0, 1]$ for each node. The recall and optimization outcomes (i.e., And Reduction) of Random for various values of $k$ are summarized in Table 8. The results reveal a near-linearly positive relationship between the value of $k$ and the recall, with a similar trend observed between the recall and the optimization performance as well. Therefore, to maintain the optimization performance of heuristics, it is essential for our model to maximize prediction recall.

### B.2  MORE RESULTS FOR COMPARATIVE EVALUATION

In this subsection, we provide further insights into the efficiency of our HIS compared to several baselines, including COG-Mfs2, CMO-Mfs2, Effisyn-Mfs2, and Random-Mfs2. To ensure a fair comparison, we select higher hyperparameter values for $k$ in the baselines, which are necessary to achieve comparable online optimization performance. While higher values of $k$ improve optimization performance, they also increase time costs, so this parameter is adjusted differently for each circuit. Specifically, we use $k = 60\%$ for COG and CMO across most circuits, and higher values of $k$ for Effisyn and Random, based on the specific circuit. The results and details are shown in Table 7. Additionally, when compared to COG-Mfs2 and CMO-Mfs2, our HIS achieves average runtime improvements of 22. 91% and 11. 96%, respectively, while maintaining or even improving the optimization performance across most circuits. Moreover, when compared to Random-Mfs2 and Effisyn-Mfs2, with $k$ values not lower than 70%, our HIS demonstrates average improvements of 19.24% and 21.82%.

### B.3  MORE RESULTS FOR INFERENCE EFFICIENCY

In this subsection, we present further details on the inference efficiency of our HIS, compared to COG, CMO, and Effisyn. Note that our HIS and COG both rely on graph inputs, which enhance optimization performance but result in lower inference efficiency. As shown in Table 5, our HIS achieves an average inference speedup of $296 \times$ on the EPFL circuits and $254\times$ on IWLS circuits. These results reveal that HIS successfully learns a lightweight graph-based symbolic scoring function, delivering both high prediction recall and efficient inference. This makes HIS well-suited for deployment in real-world industrial applications. Although CMO and Effisyn show higher inference efficiencies than HIS, their actual runtimes are comparable.

## C  DETAILS OF DATASETS USED IN THIS PAPER

### C.1  DESCRIPTION OF TWO WIDELY USED BENCHMARKS

We provide detailed statistics of the circuits from two open source benchmarks EPFL and IWLS in Tables 9 and 10, respectively. These benchmarks contain 41 circuits in total. In general, nodes

denote logic gates and edges represent the wires connecting them. The fanins refer to the nodes that provide inputs to it, while its fanouts are the nodes it drives. Primary inputs (PIs) are nodes without fanins, and primary outputs (POs) are a subset of the network's nodes. Latches are specialized nodes found in sequential circuits, and cubes denote specific subsets of input variables. Lev refers to the depth of the circuit, measured by the maximum number of edges between PIs and POs.

## C.2 DATASETS FOR EVALUATION ON OPEN-SOURCE BENCHMARKS

For each circuit and a given X heuristic, we generate the circuit dataset by applying the X heuristic to optimize the circuit, then collecting the graph features $\{\mathcal{G}_i\}_{i=1}^n$ and labels $\{y_i\}_{i=1}^n$. We observe that a few circuits contain no effective nodes, and we exclude these from our analysis since no transformations need to be applied to them, thus negating the need for model training.

In particular, employing the generalizable evaluation strategy with the EPFL benchmark, we construct three datasets for evaluation. One of the three circuit datasets——collected from Hyp, Multiplier, and Square——serves as the testing dataset, while the circuits from the IWLS are used for training. Similarly, using the generalization strategy with the IWLS benchmark, we create three datasets, selecting one of these circuit datasets from DesPerf, Ethernet, and Conmax as the testing dataset and using the EPFL circuits for training.

# D DETAILS OF METHODS AND EXPERIMENTAL SETTINGS

## D.1 DETAILS OF EXPERIMENTAL SETUP

### D.1.1 OPTIMIZATION SEQUENCE FLOWS

**Optimization Sequence Flows for Data Collection and Evaluation** In industrial practice, a sequence of Logic Optimization (LO) heuristics is typically applied to optimize an input circuit. We adopt the same setting throughout all experiments unless stated otherwise. Specifically, for the Mfs2 heuristic, we use the sequence *strash; dch; if -C 12; mfs2 -W 4 -M 5000* to collect graph data and evaluate both the Default Mfs2 heuristic and our proposed HIS. Note that the optimization sequence flow is a standard academic flow for evaluating the Default Mfs2 heuristic, which follows previous work (Mishchenko et al., 2011; Li et al., 2023; Wang et al.).

**Optimization Sequence Flows for Evaluating 2HIS-Mfs2** To apply our HIS twice, we adopt the heuristic sequence *strash; dch; if -C 12; mfs2 -W 4 -M 5000; strash; if -C 12; mfs2 -W 4 -M 5000* for evaluating the performance of 2HIS. The Mfs2 heuristic is a post-mapping optimization technique that operates on a $k$-input look-up table graph (K-LUT). Specifically, the *strash* heuristic (Rai et al., 2021) converts the circuit into an And-Inverter Graph (AIG) using one-level structural hashing, while the *if* heuristic (Mishchenko et al., 2007) maps the AIG into K-LUTs. Finally, the Mfs2 heuristic performs optimization on the resulting K-LUTs twice.

### D.1.2 TOP K ACCURACY METRIC

A common challenge in many LO heuristics is the ineffective node-level transformations problem, where the number of ineffective nodes substantially exceeds the number of effective ones. This imbalance introduces a significant distribution shift in the training dataset, making the normal threshold of 0.5 unsuitable for determining whether a sample is positive. To mitigate this issue, we adopt the approach proposed in (Wang et al.), which reformulates the classification task as a ranking problem. Specifically, all nodes are ranked according to the prediction scores assigned by the learned symbolic functions, and the top-$k$ nodes are selected as positives while the remainder are classified as negatives. The evaluation metric, referred to as top-$k$ accuracy, is defined as the proportion of true positive nodes in the top-$k$ predictions that are correctly identified, i.e., recall.

## D.2 IMPLEMENTATION DETAILS OF THE BASELINES

In this part, we present a detailed description of all the baselines used in this paper.

**COG.** COG is a well-designed 2-layer graph convolutional neural network that can achieve high optimization performance (Wang et al.). Specifically, it constructs a bipartite graph as input and learns a domain-invariant representation to achieve high generalization capability.

**CMO.** CMO is a novel graph-enhanced symbolic discovery framework (Bai et al.). Specifically, it employs a Monte Carlo method to explore the symbolic function space, while leveraging a well-designed GNN as a teacher model to guide the search process. This approach achieves state-of-the-art performance among lightweight scoring function methods.

**Effisyn**. Effisyn is a human-designed nonlinear symbolic function (Li et al., 2023). Specifically, in human-designed symbolic scoring functions, experts manually design the structure of the function and extract key parameters from training circuit data to form a complete symbolic scoring function. This process involves identifying relevant characteristics of the circuit and carefully selecting or engineering the symbolic terms that best capture the underlying behavior of the system. However, designing and developing these functions is extremely challenging as it requires extensive expert knowledge and manual tuning.

**Random.** Random is a baseline that randomly predicts a score between $[0, 1]$ for each node, and selects the top k nodes as positive samples to apply node-level transformations.

### D.3    Implementation Details of The Training Details

In this subsection, we provide further details of the training process. The overall procedure of our algorithm is illustrated in Algorithm 1, and the corresponding parameter settings are summarized in Table 4, covering model, reinforcement learning, and symbolic tree configurations. Moreover, we adopt a Best-of-$N$ (BON) strategy during training. Specifically, the top-$N$ expressions with the highest training rewards are selected to construct an ensemble model. Owing to the lightweight nature of the symbolic functions learned by HIS, the additional computational overhead is negligible, while yielding substantial performance improvements. In our experiments, we set $N = 4$.

### D.4    Implementation Details of The Circuit Subgraph Construction

In this subsection, we describe the procedure for constructing circuit subgraphs and modeling them as computation trees. Following (Wang et al.), a subgraph in LS heuristics is constructed by selecting a root node along with a limited set of its neighboring nodes. To enable more effective node embedding alignment, we first transform the subgraph into a bipartite graph, where the root node and the non-root nodes are treated as two distinct types of nodes. This bipartite graph is then converted into a two-layer computation tree: the root node corresponds to the $0$-th and $2$-th layers, while all candidate nodes in the subgraph are placed in the $1$-th layer.

## E    The Use of Large Language Model

In accordance with the ICLR 2026 policy, we disclose the use of Large Language Models (LLMs) as an assistive tool in preparing this manuscript. The primary role of LLMs was to support improvements in writing clarity and presentation quality.

Specifically, LLMs were used for the following purposes:

- **Grammar and Spelling Correction:** Detecting and correcting grammatical errors and typographical mistakes.
- **Clarity and Readability:** Rephrasing sentences and suggesting alternative formulations to enhance readability and flow.
- **Conciseness:** Streamlining sentences and paragraphs to make the writing more direct and succinct.

All scientific contributions, analyses, and claims in this paper are solely the work of the human authors. The use of LLMs was limited to language refinement and carried out responsibly in accordance with academic and ethical standards.

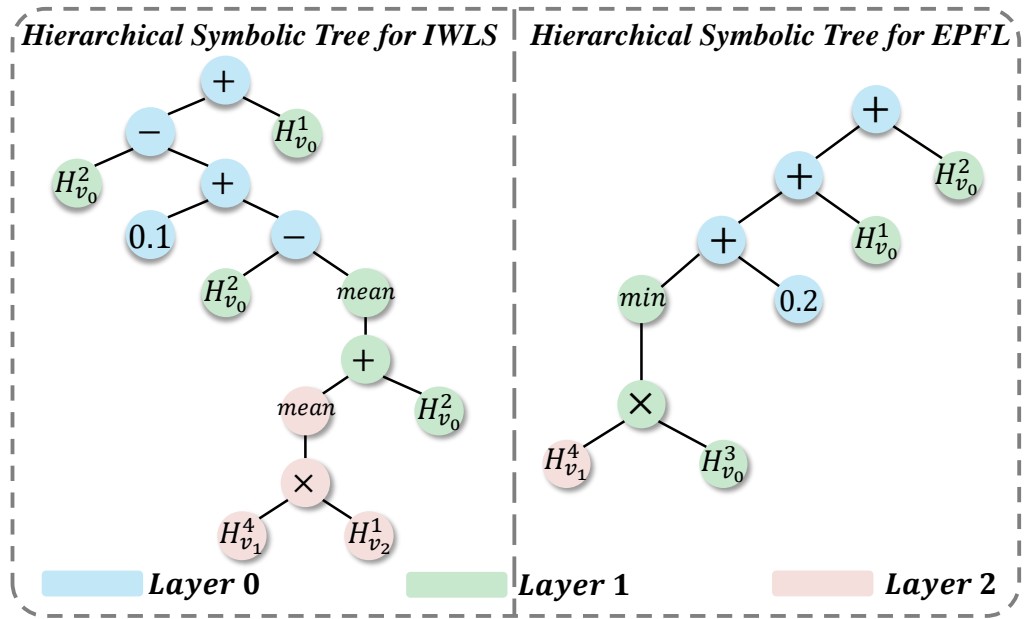

Figure 5: Visualization of the hierarchical symbolic functions for EPFL and IWLS benchmarks.

---

**Algorithm 1** Circuit Symbolic Generation Framework

---

**Input:** Transformer model $p_{\boldsymbol{\theta}}$, symbolic library $\mathcal{L}$, dataset $D = \{(T_{v_i}^L, y_i)\}_{i=1}^n$, number of generated expressions $m$, number of layers $L$

**Output:** Updated parameter $\boldsymbol{\theta}$.

  **for** $i = 1$ to training epoch **do**

    Initialize empty list of hierarchical symbolic trees

    **for** $j = 1$ to $m$ **do**

      Initialize empty sequence $\tau$

      **for** $l = 0$ down to $L$ **do**

        **while** sequence not completed **do**

          Sample a token: $\tau_k^l \sim p_{\theta_l}(\tau_k^l | \tau_1^l, \ldots, \tau_{l-1}^l)$

          Apply constraints to $\tau_k^l$

        **end while**

      **end for**

      Merge $L$ sequences $\{\tau^i\}_{i=0}^L$ into a hierarchical symbolic tree $\boldsymbol{\tau}$

      **if** $\tau$ contains updated feature tokens **then**

        Substitute feature tokens with corresponding sequences

      **end if**

      Add $\tau$ to the list of generated trees

    **end for**

    Compute rewards for each tree $\tau$:

$$r(\boldsymbol{\tau}) = -\frac{1}{n} \sum_{i=1}^n \left[ \alpha y_i (1 - \hat{y}_i)^\gamma \log(\hat{y}_i) + (1 - \alpha)(1 - y_i)\hat{y}_i^\gamma \log(1 - \hat{y}_i) \right]$$

    Update model parameters $\boldsymbol{\theta}$ using PPO objective:

$$J(\boldsymbol{\theta}) = \mathbb{E}_{\boldsymbol{\tau} \sim p(\boldsymbol{\tau}|\boldsymbol{\theta})} \left[ \min\left( \frac{p_{\boldsymbol{\theta}}(\boldsymbol{\tau})}{p_{\boldsymbol{\theta}_{\text{old}}}(\boldsymbol{\tau})} A_{\boldsymbol{\theta}_{\text{old}}}(\boldsymbol{\tau}),\ \text{clip}\left( \frac{p_{\boldsymbol{\theta}}(\boldsymbol{\tau})}{p_{\boldsymbol{\theta}_{\text{old}}}(\boldsymbol{\tau})}, 1 - \epsilon, 1 + \epsilon \right) A_{\boldsymbol{\theta}_{\text{old}}}(\boldsymbol{\tau}) \right) \right]$$

  **end for**

---

Table 4: We provide comprehensive implementation details, including the arguments for training, the Transformer model, and RL algorithms, along with a subset of the tokens library.

| Parameter | Value |
|---|---|
| Train Kwargs | |
| number of expressions generated each epoch | 512 |
| data_batch_size | 10240 |
| number of expressions for RL training | 96 |
| number of expressions for recording | 16 |
| training epoch | 2000 |
| Transformer Kwargs | |
| Transformer min_length of each layer | [4, 4, 4] |
| Transformer max_length of each layer | [48, 16, 8] |
| Transformer embedding dimension | 32 |
| Transformer attention heads | 4 |
| Transformer feedforward model dimension | 128 |
| Transformer number of layers | 4 |
| RL Kwargs | |
| PPO learning rate | 5e-5 |
| PPO epochs at each iteration | 10 |
| PPO clipping threshold | 0.2 |
| Symbolic Tree Kwargs | |
| Constant operators | [0.1, 0.2, 0.5] |
| Math operators | $\{+, -, \times, \div, \log, \exp\}$ |
| aggregation operators | $\{min, max, mean, sum\}$ |

Table 5: The model inference results show that our HIS is extremely efficient for inference compared to the SOTA graph-based approach (COG) when executed on CPU-based LO tools.

| EPFL Benchmark | Hyp | | Square | | Multiplier | | Average | |
|---|---|---|---|---|---|---|---|---|
| Method | Inference time (s) ↓ | Improvement ↑ | Inference time (s) ↓ | Improvement ↑ | Inference time (s) ↓ | Improvement ↑ | Inference time (s) ↓ | Improvement ↑ |
| COG | 5.631 | NA | 0.418 | NA | 0.865 | NA | 2.304 | NA |
| CMO | 0.003 | 2097 | 0.001 | 761 | 0.001 | 1080 | 0.001 | 1313 |
| Effisyn | 0.005 | 1228 | 0.001 | 760 | 0.001 | 1087 | 0.002 | 1025 |
| HIS (Ours) | 0.009 | 635 | 0.003 | 162 | 0.010 | 90 | 0.007 | 296 |
| **IWLS Benchmark** | DesPerf | | Ethernet | | Conmax | | **Average** | |
| Method | Inference time (s) ↓ | Improvement ↑ | Inference time (s) ↓ | Improvement ↑ | Inference time (s) ↓ | Improvement ↑ | Inference time (s) ↓ | Improvement ↑ |
| COG | 2.507 | NA | 1.050 | NA | 1.388 | NA | 1.648 | NA |
| CMO | 0.002 | 1598 | 0.000 | 2501 | 0.001 | 1370 | 0.001 | 1823 |
| Effisyn | 0.003 | 907 | 0.001 | 1304 | 0.001 | 961 | 0.002 | 1057 |
| HIS (Ours) | 0.007 | 342 | 0.005 | 197 | 0.006 | 223 | 0.006 | 254 |

Table 6: The discovered hierarchical symbolic functions for the IWLS and EPFL benchmarks.

| | | | | | | | **IWLS** | | | |
|---|---|---|---|---|---|---|---|---|---|---|
| **layer0** | Final output | | | | | | score | | | |
| | Expression | | | | | | $\left(\hat{H}_{v_0}^1 - \left(0.1 + (\hat{H}_{v_0}^1 - \hat{H}_{v_0}^9)\right)\right) + \hat{H}_{v_0}^0$ | | | |
| **layer1** | Updated Feature | $\hat{H}_{v_0}^0$ | $\hat{H}_{v_0}^1$ | $\hat{H}_{v_0}^2$ | $\hat{H}_{v_0}^3$ | $\hat{H}_{v_0}^4$ | $\hat{H}_{v_0}^5$ | $\hat{H}_{v_0}^6$ | $\hat{H}_{v_0}^7$ | $\hat{H}_{v_0}^8$ ... $\hat{H}_{v_0}^9$ |
| | Expression | $H_{v_0}^0$ | $H_{v_0}^1$ | $H_{v_0}^2$ | $H_{v_0}^3$ | $H_{v_0}^4$ | $sum(0.1 + \hat{H}_{v_1}^3)$ | $max(0.1 - \hat{H}_{v_1}^4)$ | $max(\hat{H}_{v_1}^1 + \hat{H}_{v_1}^2)$ | $min(\hat{H}_{v_1}^1 * \hat{H}_{v_1}^3)$ ... $mean(\hat{H}_{v_1}^5 + H_{v_0}^1)$ |
| **layer2** | Updated Feature | $\hat{H}_{v_1}^0$ | $\hat{H}_{v_1}^1$ | $\hat{H}_{v_1}^2$ | $\hat{H}_{v_1}^3$ | $\hat{H}_{v_1}^4$ | $\hat{H}_{v_1}^5$ | $\hat{H}_{v_1}^6$ | $\hat{H}_{v_1}^7$ | $\hat{H}_{v_1}^8$ ... $\hat{H}_{v_1}^9$ |
| | Expression | $H_{v_1}^0$ | $H_{v_1}^1$ | $H_{v_1}^2$ | $H_{v_1}^3$ | $H_{v_1}^4$ | $mean(H_{v_1}^3 * H_{v_2}^0)$ | $min(H_{v_1}^3 + H_{v_2}^1)$ | $mean(H_{v_1}^3 + H_{v_2}^2)$ | $mean(H_{v_1}^3 - H_{v_2}^0)$ ... $min(H_{v_2}^1 + H_{v_2}^0)$ |
| | | | | | | | **EPFL** | | | |
| **layer0** | Final output | | | | | | score | | | |
| | Expression | | | | | | $\left(((\hat{H}_{v_0}^6 + 0.2) + \hat{H}_{v_0}^0) + \hat{H}_{v_0}^1\right)$ | | | |
| **layer1** | Updated Feature | $\hat{H}_{v_0}^0$ | $\hat{H}_{v_0}^1$ | $\hat{H}_{v_0}^2$ | $\hat{H}_{v_0}^3$ | $\hat{H}_{v_0}^4$ | $\hat{H}_{v_0}^5$ | $\hat{H}_{v_0}^6$ | $\hat{H}_{v_0}^7$ | $\hat{H}_{v_0}^8$ ... $\hat{H}_{v_0}^9$ |
| | Expression | $H_{v_0}^0$ | $H_{v_0}^1$ | $H_{v_0}^2$ | $H_{v_0}^3$ | $H_{v_0}^4$ | $min(\hat{H}_{v_1}^3 - 0.5)$ | $min(\hat{H}_{v_1}^3 * H_{v_0}^2)$ | $max(\hat{H}_{v_1}^4 - \hat{H}_{v_1}^2)$ | $min(0.5 * \hat{H}_{v_1}^2)$ ... $min(H_{v_0}^1 + \hat{H}_{v_1}^1)$ |
| **layer2** | Updated Feature | $\hat{H}_{v_1}^0$ | $\hat{H}_{v_1}^1$ | $\hat{H}_{v_1}^2$ | $\hat{H}_{v_1}^3$ | $\hat{H}_{v_1}^4$ | $\hat{H}_{v_1}^5$ | $\hat{H}_{v_1}^6$ | $\hat{H}_{v_1}^7$ | $\hat{H}_{v_1}^8$ ... $\hat{H}_{v_1}^9$ |
| | Expression | $H_{v_1}^0$ | $H_{v_1}^1$ | $H_{v_1}^2$ | $H_{v_1}^3$ | $H_{v_1}^4$ | $max(H_{v_1}^2 + H_{v_2}^4)$ | $max(H_{v_2}^1 + 0.2)$ | $min(0.1 * H_{v_2}^1)$ | $min(0.2 - H_{v_2}^1)$ ... $mean(H_{v_2}^3 + H_{v_1}^2)$ |

Table 7: We compare our HIS with four competitive baselines. The results demonstrate that our approach consistently outperforms all baselines in terms of online heuristics efficiency and optimization performance. And Reduction (AR) denotes the reduced number of nodes, i.e., optimization performance. Normalized AR denotes the ratio of the AR to that of the default heuristic.

| **Hyp** | | | | **Square** | | | |
|---|---|---|---|---|---|---|---|
| Method | And Reduction(AR) ↑ | Normalized AR ↑ | Times(s) ↓ | Method | And Reduction(AR) ↑ | Normalized AR ↑ | Times(s) ↓ |
| COG | 435 | 0.66 | 198.09 | COG | 6 | 0.75 | 14.70 |
| CMO | 142 | 0.21 | 129.56 | CMO | 6 | 0.75 | 13.60 |
| Random | 563 | 0.85 | 238.22 | Random | 3 | 0.38 | 15.62 |
| Effisyn | 498 | 0.75 | 218.20 | Effisyn | 3 | 0.38 | 14.66 |
| HIS (Ours) | 566 | **0.85** | **85.46** | HIS (Ours) | 6 | **0.75** | **10.69** |
| **Multiplier** | | | | **DesPerf** | | | |
| Method | And Reduction(AR) ↑ | Normalized AR ↑ | Times(s) ↓ | Method | And Reduction(AR) ↑ | Normalized AR ↑ | Times(s) ↓ |
| COG | 21 | 0.95 | 17.07 | COG | 732 | 0.65 | 29.01 |
| CMO | 22 | 1.00 | 15.08 | CMO | 900 | 0.81 | 22.82 |
| Random | 18 | 0.82 | 14.99 | Random | 906 | 0.81 | 23.30 |
| Effisyn | 22 | 1.00 | 15.51 | Effisyn | 886 | 0.79 | 24.68 |
| HIS (Ours) | 22 | **1.00** | **13.52** | HIS (Ours) | 936 | **0.84** | **22.84** |
| **Ethernet** | | | | **Conmax** | | | |
| Method | And Reduction(AR) ↑ | Normalized AR ↑ | Times(s) ↓ | Method | And Reduction(AR) ↑ | Normalized AR ↑ | Times(s) ↓ |
| COG | 36 | 0.95 | 18.30 | COG | 259 | 0.33 | 16.66 |
| CMO | 2 | 0.05 | 20.88 | CMO | 730 | **0.93** | 14.43 |
| Random | 19 | 0.50 | 19.04 | Random | 579 | 0.74 | 14.25 |
| Effisyn | 24 | 0.63 | 20.93 | Effisyn | 593 | 0.76 | 16.76 |
| HIS (Ours) | 37 | **0.97** | **13.19** | HIS (Ours) | 647 | 0.83 | **13.63** |

Table 8: We report the recall and optimization performance of the Mfs2 heuristic incorporated with Random models. Percent denotes the hyperparameter k, i.e., the percent of nodes to apply transformations. And Reduction denotes the reduced number of nodes, i.e., optimization performance.

| Hyp | | | Multiplier | | |
|---|---|---|---|---|---|
| Percent | Recall | And Reduction(AR) | Percent | Recall | And Reduction(AR) |
| 0.10 | 0.11 | 33.33 | 0.10 | 0.10 | 3.00 |
| 0.20 | 0.20 | 69.00 | 0.20 | 0.18 | 5.33 |
| 0.30 | 0.30 | 111.33 | 0.30 | 0.28 | 6.67 |
| 0.40 | 0.40 | 164.67 | 0.40 | 0.39 | 9.33 |
| 0.50 | 0.50 | 225.33 | 0.50 | 0.44 | 10.00 |
| 0.60 | 0.60 | 295.00 | 0.60 | 0.56 | 12.33 |
| 0.70 | 0.70 | 374.33 | 0.70 | 0.67 | 14.00 |
| 0.80 | 0.80 | 464.33 | 0.80 | 0.78 | 16.67 |
| 0.90 | 0.90 | 561.33 | 0.90 | 0.89 | 19.00 |
| 1.00 | 1.00 | 664.00 | 1.00 | 1.00 | 22.00 |
| Square | | | DesPerf | | |
| Percent | Recall | And Reduction(AR) | Percent | Recall | And Reduction(AR) |
| 0.10 | 0.10 | 114.67 | 0.10 | 0.10 | 114.67 |
| 0.20 | 0.21 | 210.33 | 0.20 | 0.21 | 210.33 |
| 0.30 | 0.31 | 318.33 | 0.30 | 0.31 | 318.33 |
| 0.40 | 0.41 | 421.33 | 0.40 | 0.41 | 421.33 |
| 0.50 | 0.50 | 529.67 | 0.50 | 0.50 | 529.67 |
| 0.60 | 0.60 | 657.67 | 0.60 | 0.60 | 657.67 |
| 0.70 | 0.70 | 790.00 | 0.70 | 0.70 | 790.00 |
| 0.80 | 0.80 | 904.67 | 0.80 | 0.80 | 904.67 |
| 0.90 | 0.90 | 1001.33 | 0.90 | 0.90 | 1001.33 |
| 1.00 | 1.00 | 1118.00 | 1.00 | 1.00 | 1118.00 |
| Ethernet | | | Conmax | | |
| Percent | Recall | And Reduction(AR) | Percent | Recall | And Reduction(AR) |
| 0.10 | 0.11 | 0.00 | 0.10 | 0.10 | 95.00 |
| 0.20 | 0.19 | 0.00 | 0.20 | 0.20 | 188.00 |
| 0.30 | 0.28 | 0.33 | 0.30 | 0.30 | 251.00 |
| 0.40 | 0.38 | 1.33 | 0.40 | 0.40 | 330.67 |
| 0.50 | 0.48 | 2.33 | 0.50 | 0.50 | 411.67 |
| 0.60 | 0.56 | 3.00 | 0.60 | 0.59 | 493.33 |
| 0.70 | 0.65 | 3.67 | 0.70 | 0.69 | 557.67 |
| 0.80 | 0.75 | 4.33 | 0.80 | 0.78 | 625.00 |
| 0.90 | 0.89 | 6.67 | 0.90 | 0.90 | 718.67 |
| 1.00 | 1.00 | 8.00 | 1.00 | 1.00 | 782.00 |

Table 9: A detailed description of circuits from the EPFL benchmark. Nodes denotes the sizes of circuits, and Lev denotes the depths of circuits.

| Circuit | PI | PO | Latch | Nodes | Edge | Cube | Lev |
|---|---|---|---|---|---|---|---|
| Adder | 256 | 129 | 0 | 1020 | 2040 | 1020 | 255 |
| Barrel shifter | 135 | 128 | 0 | 3336 | 6672 | 3336 | 12 |
| Divisor | 128 | 128 | 0 | 57247 | 114494 | 57247 | 4372 |
| Hypotenuse | 256 | 128 | 0 | 214335 | 428670 | 214335 | 24801 |
| Log2 | 32 | 32 | 0 | 32060 | 64120 | 323060 | 444 |
| Max | 512 | 130 | 0 | 2865 | 5730 | 2865 | 287 |
| Multiplier | 128 | 128 | 0 | 27062 | 54124 | 27062 | 274 |
| Sin | 24 | 25 | 0 | 5416 | 10832 | 5416 | 225 |
| Square-root | 128 | 64 | 0 | 24618 | 49236 | 24618 | 5058 |
| Square | 64 | 128 | 0 | 18486 | 36969 | 18485 | 250 |
| Round-robin ariter | 256 | 129 | 0 | 11839 | 23678 | 11839 | 87 |
| Alu control unit | 7 | 26 | 0 | 175 | 348 | 174 | 10 |
| Coding-cavlc | 10 | 11 | 0 | 693 | 1386 | 693 | 16 |
| Decoder | 8 | 256 | 0 | 304 | 608 | 304 | 3 |
| i2c controller | 147 | 142 | 0 | 1357 | 2698 | 1356 | 20 |
| Int to float converter | 11 | 7 | 0 | 260 | 520 | 260 | 16 |
| Memory controller | 1204 | 1230 | 0 | 47110 | 93945 | 47109 | 114 |
| Priority encoder | 128 | 8 | 0 | 978 | 1956 | 978 | 250 |
| Lookahead XY router | 60 | 30 | 0 | 284 | 514 | 257 | 54 |
| Voter | 1001 | 1 | 0 | 13758 | 27516 | 13758 | 70 |

Table 10: A detailed description of circuits from the IWLS benchmark. Nodes denotes the sizes of circuits, and Lev denotes the depths of circuits.

| Circuit | PI | PO | latch | nodes | edge | cube | lev |
|---|---|---|---|---|---|---|---|
| aes_core | 259 | 129 | 530 | 20797 | 40645 | 24444 | 28 |
| des_area | 240 | 64 | 128 | 5005 | 9882 | 5889 | 35 |
| des_perf | 234 | 64 | 8808 | 98463 | 180542 | 108666 | 28 |
| ethernet | 98 | 115 | 10544 | 46804 | 113378 | 72850 | 37 |
| i2c | 19 | 14 | 128 | 1147 | 2299 | 1375 | 15 |
| mem_ctrl | 115 | 152 | 1083 | 11508 | 26436 | 14603 | 31 |
| pci_bridge32 | 162 | 207 | 3359 | 16897 | 34607 | 23130 | 29 |
| pci_conf_cyc_addr_dec | 32 | 32 | 0 | 109 | 212 | 128 | 6 |
| pci_spoci_ctrl | 25 | 13 | 60 | 1271 | 2637 | 1557 | 19 |
| sasc | 16 | 12 | 117 | 552 | 1148 | 766 | 10 |
| simple_spi | 16 | 12 | 132 | 823 | 1694 | 1089 | 14 |
| spi | 47 | 45 | 229 | 3230 | 6904 | 4054 | 32 |
| steppermotordrive | 4 | 4 | 25 | 228 | 397 | 253 | 11 |
| systemcaes | 260 | 129 | 670 | 7961 | 18236 | 11648 | 44 |
| systemcdes | 132 | 65 | 190 | 3324 | 6304 | 3791 | 33 |
| tv80 | 14 | 32 | 359 | 7166 | 16280 | 9352 | 50 |
| usb_funct | 128 | 121 | 1746 | 12871 | 27102 | 16378 | 25 |
| usb_phy | 15 | 18 | 98 | 559 | 1001 | 638 | 12 |
| vga_lcd | 89 | 109 | 17079 | 124050 | 242332 | 146201 | 25 |
| wb_conmax | 1130 | 1416 | 770 | 29036 | 77185 | 39619 | 26 |
| wb_dma | 217 | 215 | 263 | 3495 | 7052 | 4496 | 26 |

