# OpenReview forum: "A Hierarchical Circuit Symbolic Discovery Framework for Efficient Logic Optimization"
_ICLR.cc/2026/Conference — ICLR 2026 Poster_

### Official Review · Reviewer_KPds · 2025-10-28

**Soundness:** 2
**Presentation:** 2
**Contribution:** 2
**Rating:** 4
**Confidence:** 4

**Summary:**

To address the bottleneck of low efficiency in LO heuristic algorithms for chip design, as well as the issues of high inference cost and poor interpretability in existing GNN-based optimization methods, the researchers propose HIS, a hierarchical circuit symbol discovery framework. This framework leverages a hierarchical symbolic tree structure and a structure-aware Transformer optimized with PPO to generate lightweight, interpretable symbolic functions for identifying invalid subgraphs. Evaluated on the EPFL and IWLS benchmarks, HIS outperforms SOTA methods like COG and CMO in terms of prediction recall. Furthermore, when integrated with Mfs2, it achieves an average runtime reduction of 27.22%, a circuit size reduction of 6.95%, and demonstrates 296x and 254x faster inference speed compared to COG.​

**Strengths:**

1. The researchers tackle the critical problem of logic optimization in EDA, achieving notable advances on an NP-hard problem through the design of HIS.
2. The study validates the balance between efficiency, accuracy, and interpretability on industrial-grade benchmarks. The experimental design is solid and demonstrates high reproducibility, achieving a 27.22% reduction in runtime and a 6.95% reduction in circuit size.

**Weaknesses:**

1. The work lacks a mathematical proof that the hierarchical symbolic tree can equivalently approximate the message-passing capability of a GNN. It relies solely on the experimental observation that its performance with L=2 is comparable to that of a 2-layer GNN. The work lacks a theoretical analysis of the necessity of the operators (e.g., log, exp) included in the symbol library.
2. The paper does not derive the generalization error bound for the symbolic function, such as how the sensitivity of the $\beta / \gamma$ parameters changes as the number of circuit nodes increases. While positioning itself as a paper that emphasizes "interpretability," it provides little theoretical proof or analysis to substantiate this claim.
3. This paper only validates the Mfs2 heuristic for logic optimization, without testing mainstream pre-mapping heuristics such as Resub and Rewrite. In my view, the practical value of this work in the broader context of LO problems would be significantly strengthened if it included at least three different LO heuristics. Otherwise, there is reasonable doubt that the proposed framework might be effective only for Mfs2.
4. While the EPFL and IWLS benchmarks cover industrial-scale circuits, they do not include real circuit data from advanced process technologies (e.g., 7nm or 3nm). Therefore, I believe testing on real industrial data would provide more compelling evidence of the method's robustness.
5. The work fails to investigate the impact of the computation tree depth L (using only L=2) on performance, thus making it impossible to determine if L=3 or L=4 could enhance the structure-capturing capability. It also omits an analysis of how circuit type (e.g., combinational vs. sequential logic) influences the effectiveness of HIS.
6. It is recommended to include a comparison with more recent methods like Boolformer.

**Questions:**

1. Please respond to the concerns I have raised in the 'weaknesses' section. If the revision can adequately address most of the critical issues, I would consider raising the score.
2. In my personal opinion, the contribution of this paper lies more in its solid experimentation and reproducibility than in its theoretical advancement in ML. Considering this, EDA-focused conferences (such as DAC or ISCA, with an upcoming November deadline) would be a more suitable venue for it. Of course, I also believe that if the intention is to submit to DAC or ISCA, experiments related to scalability, as well as validation under advanced process technologies (below 7nm), would need to be supplemented.

---

> ### Author Response · Authors · 2025-11-22
> **Response to Reviewer KPds---Part1/5**
>
> Dear Reviewer KPds,
>
> We greatly appreciate your careful reading and constructive feedback! We have provided further responses as follows. We sincerely hope that our responses could properly address all your concerns. If so, we would deeply appreciate it if you could raise your score. If not, please let us know your further concerns, and we will continue actively responding to your comments and improving our submission.
>
> ### Weakness 1.1
> >**The work lacks a mathematical proof that the hierarchical symbolic tree can equivalently approximate the message-passing capability of a GNN.**
>
> We sincerely appreciate the reviewer for pointing out this important problem. Following your suggestion, we have added a complete mathematical proof (**see Theorem 1 and its corresponding proof in the revised version**) demonstrating that the proposed hierarchical symbolic tree can equivalently approximate the message-passing capability of a GNN under mild assumptions. For your convenience, we provide the detailed theorem as follows.
>
> **Theorem 1 (Uniform Approximation of GNN Message Passing).**  Consider a depth-$L$ ($L>0$) message-passing tree $T_v^L$, whose GNN computation is $y_{\mathrm{GNN}}(T_v^L) = \Phi_0 \circ \Phi_1 \circ \cdots \circ \Phi_L(H_{v_L}, H_{v_{L-1}})$, where $H_{v_i}$ denotes the node feature matrix at layer $i$. For each $l = 0, \dots, L$, let $K_l \subset \mathbb{R}^{d_l}$ be the compact domain of input features before the $l$-th message-passing, $\hat{K_l} \subset \mathbb{R}^{\hat{d_l}}$ be the compact domain after the update, $\mathcal{A_l}$ be the symbolic class, and $C(K_l)$ be the space of continuous real-valued functions on $K_l$. Assume that **(i)** The GNN map $\Phi_l$ is continuous and $L_l$-Lipschitz on its compact domain, and **(ii)** the symbolic class $\mathcal{A_l}$ is a subalgebra containing basic operators (i.e., $\{+, -, \times, \div\}$) and constants, hence uniformly dense in $C(K_l)$ by **the Stone--Weierstrass theorem [1]**. Then, for any $\varepsilon > 0$, there exist functions $F_l \in \mathcal{A_l}$ such that the hierarchical symbolic function $y_{\mathrm{HIS}}(T_v^L) = F_0 \circ F_1 \circ \cdots \circ F_L(H_{v_L}, H_{v_{L-1}})$ satisfies the error bound
> $$
> \sup_{(H_{v_L}, H_{v_{L-1}})} \big| y_{\mathrm{GNN}}(T_v^L) - y_{\mathrm{HIS}}(T_v^L) \big|
> \le \sum_{l=0}^{L} \left( \delta_l \prod_{t=0}^{l-1} L_t \right),
> $$
> where $\delta_l = \sup_{z=(H_{v_L}, H_{v_{L-1}})} \| \Phi_l(z) - F_l(z) \| \le \varepsilon$, and the empty product $\prod_{t=0}^{-1} L_t$ is defined as $1$.
>
> Since $\epsilon$ can be made arbitrarily small, **the error bound can approach zero, demonstrating the existence of a hierarchical symbolic function that uniformly approximates the GNN message-passing process**.
>
> [1]. De Branges L. The stone-weierstrass theorem. Proceedings of the American Mathematical Society.
>
> ### Weakness 1.2
> > **The work lacks a theoretical analysis of the necessity of the complex operators (e.g., exp, log) included in the symbol library.**
>
> We sincerely appreciate the reviewer for the insightful comments. Generally, the operators in our symbol library can be grouped into two categories: **basic operators** {$+, -, \times, \div$} and **complex operators** {$exp, \log, \min, \max, \mathrm{sum}, \mathrm{mean}$}. While the basic operators are theoretically sufficient to approximate the GNN’s computation as established in Theorem 1, **the complex operators are essential for achieving efficient approximation of the nonlinear transformations** frequently used in modern GNN message-passing mechanisms.
>
> Specifically, many functions $f$ in modern GNNs—such as **softmax-like normalization** and **message aggregation**—critically rely on complex operators. Although polynomials $p$ constructed from basic operators are theoretically sufficient to approximate these functions, classical approximation results [2] demonstrate that the minimal polynomial degree
> $$
> d = \min\\{\deg(p) : \\|p-f\\|_{L\_\infty(K)} < \\varepsilon\\}
> $$
> must satisfy strong lower approximation bounds:
> - $d = \Omega(a)$ for approximating **$\exp(x)$** on $[-a,a]$,
> - $d = \Omega(1/c)$ for approximating **$\log(x)$** on $[c,1]$, $c > 0$.
>
> These results indicate that accurate approximation of functions such as $\exp$ and $\log$ inevitably requires high-degree polynomials. Consequently, **restricting the hierarchical symbolic tree to polynomial (basic) operators causes the expression depth and width to grow rapidly**, undermining interpretability and substantially increasing the search complexity. Therefore, the complex operators are necessary in our symbol libray to achieve efficient approximation.
>
> [2]. Aggarwal A, Alman J. Optimal-Degree Polynomial Approximations for Exponentials and Gaussian Kernel Density Estimation. Computational Complexity Conference, 2022.

---

> ### Author Response · Authors · 2025-11-22
> **Response to Reviewer KPds---Part2/5**
>
> ### Weakness 2
> >**The paper does not derive the generalization error bound for the symbolic function, such as how the sensitivity of the $\beta/\gamma$ parameters changes as the number of circuit nodes increases.**
>
> We sincerely appreciate the reviewer for the insightful comments. Regarding the generalization bound, **our HIS achieves the same bound as the GNN-based method, COG[3], due to our hierarchical symbolic function's uniform approximation of GNN as guaranteed by Theorem 1 (see Weakness 1.1)**. The generalization bound takes the form of:
> \\begin{align}
>         & \\left(\\sup \_{f \\in B_{\\bar{K}}\\left(r\\right)}\\lvert \\mathcal{R}(f)-\\hat{\\mathcal{R}}(f) \\rvert \\right)^2 \\leq \\vphantom{\\frac{C_4}{M^2}\\sum_{k=1}^M\\frac{1}{n_k}}\\frac{C_1 \\log \\delta^{-1}+C_2}{M} + \\frac{C_3\\log 2\\delta^{-1}M+C_4\\log \\delta^{-1}+C_5}{M^2}\\sum_{k=1}^M\\frac{1}{n_k}
> \\end{align}
> where $\mathcal{R}(f)$ is the average risk over all possible target Circuit Domains, $\hat{\mathcal{R}}(f)$ is its empirical risk estimation, $C_1, C_2, C_3, C_4, C_5$ are constants, $B_{\bar{K}}(r)$ denotes the ball of radius $r$ of an Reproducing Kernel Hilbert Space (RKHS) $\mathcal{H}_{\bar{K}}$, $M$ is the number of training Circuit Domains and $n_k$ is the sample size of the $k$-th Circuit Domain.
>
> Regarding the changes of the parameters sensitivity, the $\beta$ parameter refer to the tree-structured encoding vectors $\mathrm{Parent}(\tau_{p_k}^i) = \beta_p$ and $\mathrm{Sibling}(\tau_{s_k}^i) = \beta_s$.  Moreover, $\gamma$ is a constant parameter, which we set to 1 in all our experiments. **Since these parameters are learnable or constants, their precise effect on generalization bounds is difficult to analyze theoretically**. However, we can provide an empirical analysis that as the number of circuit nodes increases, the symbolic expressions become more complex, and each $\beta$ parameter is applied more frequently. Consequently, **the cumulative impact of $\beta/\gamma$ on the generalization bound will grow with the circuit size.**
>
> [3]. Wang Z, Chen L, Wang J, et al. A Circuit Domain Generalization Framework for Efficient Logic Synthesis in Chip Design. ICML, 2024.
>
> ### Weakness 3
> >**This paper only validates the Mfs2 heuristic for logic optimization, without testing mainstream pre-mapping heuristics such as Resub and Rewrite.**
>
> We sincerely appreciate the reviewer’s insightful comments. Following your suggestions, we **extended our evaluation to mainstream pre-mapping heuristics**, and the results indicate that our method generalizes well to pre-mapping stage.
>
> We fully acknowledge the reviewer's valuable comments to evaluate our method on at least three LO heuristics. However, due to the limited rebuttal period, as well as the time required for operator-specific data collection and model training, we conducted experiments on **one of the most challenging and time-consuming heuristics---Resub---as shown in Table a**. Moreover, since commonly used pre-mapping heuristics such as **Resub, Rewrite, and Refactor follow the same overall paradigm** and differ primarily in their node-level transformation rules, our approach is naturally applicable to Rewrite as well. **We will continue evaluating our method on the Rewrite heuristic and include the complete results in the camera-ready version.**
>
> Specifically, we collect training data using the same pipeline as in the Mfs2 setting in the original submission: node features are extracted from the AIG, and node-level labels are obtained based on the effectiveness of Resub transformations. We then apply our HIS method to learn symbolic scoring functions. The results in Table b demonstrate that our method achieves the best performance on most of the circuits, **with an average prediction recall of 89%, surpassing other methods**. Moreover, the online results in Table c demonstrate that our method achieves **an average runtime reduction of 36.23% with only a marginal 0.40% degradation in circuit size** across the six test circuits. Therefore, we can conclude that our HIS **generalizes effectively to both pre-mapping heuristics (e.g., Resub) and post-mapping heuristics (e.g., Mfs2).**

---

> ### Author Response · Authors · 2025-11-22
> **Response to Reviewer KPds---Part3/5**
>
> **(continued part from "Response to Reviewer KPds---Part2/5")**
>
> **Table a:** We analyze the runtime of commonly used pre-mapping heuristics on six test circuits. Ratio denotes the ratio of average runtime to that of the Rewrite heuristic. Here $K$ represents the subgraph's relative size.
> | Avg Time Ratio to Rewrite |         |         |          |       |       |    |
> | ------------------------- | ------- | ------- | -------- | ----- | ----- | ----- |
> | **Heuristics**                | Rewrite | Balance | Refactor | Resub (K=12) | Resub (K=16) |Mfs2  |
> | **Time Ratio**                | 1       | 0.05    | 1.21     | **6.01** | **73.44** | 30.94 |
>
> **Table b:** The offline results show that HIS outperforms all baselines in terms of average generalization recall **for the Resub heuristic**.
> |            | Hyp      | Square   | Multiplier | Desperf  | Ethernet | Conmax   | Average  |
> | ---------- | -------- | -------- | ---------- | -------- | -------- | -------- | -------- |
> | **Method** | Recall   | Recall   | Recall     | Recall   | Recall   | Recall   | Recall   |
> | COG        | 0.87     | 0.89     | 0.87       | 0.72 | 0.92     | 0.76     | 0.84     |
> | Effisyn    | 0.67     | 0.63     | 0.20       | 0.46     | 0.82     | 0.02     | 0.47     |
> | Random     | 0.50     | 0.48     | 0.43       | 0.51     | 0.58     | 0.49     | 0.50     |
> | CMO        | 0.90     | 0.82     | 0.88       | 0.65     | 0.91     | 0.92     | 0.85     |
> | HIS（Ours）        | 0.92 | 0.93 | 0.93   | 0.68     | 0.93 | 0.96 | **0.89** |
>
> **Table c:** We compare the Default Resub heuristic with our HIS-Resub heuristic with the hyperparameter $k$ set $50\%$ on six challenging circuits.
> |             | **Hyp**      |                       |          |                         | **Multiplier** |                       |          |                         |
> | ----------- | ------------ | --------------------- | -------- | ----------------------- | -------------- | --------------------- | -------- | ----------------------- |
> | Method      | Nd           | Nd Impr (%)$\uparrow$ | Time (s) | Time Impr (%)$\uparrow$ | Nd             | Nd Impr (%)$\uparrow$ | Time (s) | Time Impr (%)$\uparrow$ |
> | The Default | 204533.00    | NA                    | 161.82   | NA                      | 24436.00       | NA                    | 33.47    | NA                      |
> | HIS-Resub   | 205257.00    | **-0.34**             | 67.16    | **58.50**               | 24453.00       | **-0.07**                 | 25.07    | **25.11**               |
> |             | **Square**   |                       |          |                         | **DesPerf**    |                       |          |                         |
> | Method      | Nd           | Nd Impr (%)$\uparrow$ | Time (s) | Time Impr (%)$\uparrow$ | Nd             | Nd Impr (%)$\uparrow$ | Time (s) | Time Impr (%)$\uparrow$ |
> | The Default | 15845.00     | NA                    | 14.31    | NA                      | 67193.00       | NA                    | 302.69   | NA                      |
> | HIS-Resub   | 15983.00     | **-0.83**             | 7.91     | **44.73**               | 67892.00       | **-1.01**             | 242.93   | **19.74**               |
> |             | **Ethernet** |                       |          |                         | **Conmax**     |                       |          |                         |
> | Method      | Nd           | Nd Impr (%)$\uparrow$ | Time (s) | Time Impr (%)$\uparrow$ | Nd             | Nd Impr (%)$\uparrow$ | Time (s) | Time Impr (%)$\uparrow$ |
> | The Default | 43345.00     | NA                    | 109.94   | NA                      | 39126.00       | NA                    | 156.54   | NA                      |
> | HIS-Resub   | 43345.00     | **0.00**              | 82.65    | **45.06**               | 39195.00       | **-0.17**             | 118.59   | **24.24**               |

---

> ### Author Response · Authors · 2025-11-22
> **Response to Reviewer KPds---Part4/5**
>
> ### Weakness 4
> >**Lacking testing on real industrial data.**
>
> We sincerely appreciate the reviewer for the insightful comments. Following the suggestion, we have conducted **additional offline evaluations on real industrial circuits** to further assess the robustness and practical applicability of our HIS framework. Specifically, we selected **an industrial circuit benchmark from an anonymous semiconductor company**. As shown in Table d, the industrial benchmark consist of 27 circuits, where the circuit sizes range from 2775 to 788,288, significantly exceeding the complexity of open-source circuits.
>
> To assess our method on industrial-scale circuits, we **selected the four most time-consuming circuits as test cases and used the remaining circuits for training**. As shown in Table e, our approach exhibits strong generalization capability across these industrial circuits. Specifically, **HIS achieves an average prediction recall of 94%, surpassing other methods**. These results demonstrate that HIS can effectively generalize to modern industrial circuits, underscoring its practical applicability within advanced-node design flows.
>
> **Table d:** We provide detailed description of the 27 industrial circuits (including 23 training and 4 testing circuits). Here Nodes denotes the sizes of the circuits, and Lev denotes the depths of the circuits.
> | Circuit Type      | Metric | PI       | PO      | Latch | Nodes    | Lev    |
> | ----------------- | ------ | -------- | ------- | ----- | -------- | ------ |
> | Training Circuits | mean   | 8410   | 5978 | 0     | 104229 | 55  |
> |                   | max    | 59974    | 29721   | 0     | 788288   | 104    |
> |                   | min    | 41       | 107     | 0     | 2775     | 18     |
> | Testing Circuits  | mean   | 18540 | 18015   | 0     | 356111 | 103 |
> |                   | max    | 42257    | 33849   | 0     | 655243   | 185    |
> |                   | min    | 523      | 483     | 0     | 24778    | 40
>
> **Table e:**  The offline results on four challenging industrial circuits show that HIS outperforms all baselines in terms of average generalization recall for the Mfs2 heuristic.
> |  | Industrial_Ci1   | Industrial_Ci2    | Industrial_Ci3    | Industrial_Ci4    | Average  |
> | ------------------ | -------- | -------- | -------- | -------- | -------- |
> | Method             | Recall    | Recall    | Recall    | Recall    | Recall    |
> | COG                | 0.89     | 0.85     | 0.99     | 0.94     | 0.92     |
> | Effisyn            | 0.66     | 0.65     | 0.71     | 0.93     | 0.74     |
> | Random             | 0.47     | 0.52     | 0.49     | 0.56     | 0.51     |
> | CMO                | 0.79     | 0.91     | 0.98     | 0.82     | 0.88     |
> | HIS (Ours)         | **0.89** | **0.92** | **0.99** | **0.97** | **0.94** |
> ### Weakness 5
> >**Lacking investigation of the impact of the computation tree depth L (using only L=2) on performance, and an analysis of how circuit type (e.g., combinational vs. sequential logic) influences the effectiveness of HIS.**
>
> We sincerely thank the reivewer for the insightful suggestions. Below we summarize the experiments we performed and the conclusions we draw.
>
> **The impact of the computation tree depth:** To assess how the computation tree depth influences model performance, we vary the depth $L$ from 1 to 3. As shown in Table f, **HIS achieves the highest prediction performance when $L = 2$**. One possible reason for this observation is that **a shallow tree ($L = 1$) lacks sufficient expressive capacity** to approximate the underlying message-passing behavior, whereas **an excessively deep tree ($L = 3$) may introduce unnecessary complexity and lead to overfitting**. Therefore, a moderate depth strikes the best balance between expressiveness and generalization.
>
> **The impact of the circuit type:** We would like to respectively clarify that **our paper majorly focus on the LO heuristics that are applicable only to combinational circuits**. As these operators are not designed for sequential logic, it is not feasible to directly assess the influence of sequential circuit structures within our current framework. Nevertheless, we acknowledge that some logic optimization heuristics specifically target sequential circuits, and extending HIS to sequential settings represents a valuable direction for future work.
>
> **Table f:** We evaluate the impact of different computation tree depth on the prediction performance.
> |                | EPFL       |                       |    IWLS    |                       |
> | :------------: | ---------- | --------------------- | :--------: | :-------------------: |
> | Tree Depth | Recall | Training Time (h) | Recall | Training Time (h) |
> |       1        | 0.88       | 12.70                 |    0.79    |         13.27         |
> |       2        | **0.90**   | 13.76                 |  **0.86**  |         15.37         |
> |       3        | 0.88       | 15.20                 |    0.86    |         21.92         |

---

> > ### Author Response · Authors · 2025-11-22
> > **Response to Reviewer KPds---Part5/5**
> >
> > ### Weakness 6
> > >**It is recommended to include a comparison with more recent methods like Boolformer.**
> >
> > We sincerely appreciate the reviewer for the thoughtful and constructive comments. Following your suggestion, we have incorporated **an extensive comparison with some recent methods**. Specifically, the baselines include **two traditional lightweight machine learning models**---RidgeLR[4], XGBoost[5]---and four powerful symbolic regression methods---Boolformer[6], GPLearn[7], DSR[8], SPL[9]. The offline results in Table g demonstrate that **our method significantly outperforms all baselines in terms of the prediction recall**, indicating the effectiveness of our method.
> >
> > **Table g:** We compare our method with more baselines in terms of the offline prediction recall.
> > |  | Hyp      | Multiplier | Square   | DesPerf | Ethernet | Conmax | Average |
> > | ------------------ | -------- | ---------- | -------- | -------- | -------- | --------- | --------- |
> > | Method             | Recall    | Recall      | Recall    | Recall    | Recall    | Recall     | Recall     |
> > | RidgeLR    | 0.74 | 0.62 | 0.88  | 0.79 | 0.33 | 0.54  | 0.65 |
> > | XGBoost    | 0.71 | 0.86 | 0.46  | 0.79 | 0.33 | 0.68  | 0.64 |
> > | Boolformer | 0.35 | 0.65 | 0.80  | 0.65 | 0.69 | 0.62  | 0.63 |
> > | GPLearn    | 0.65 | 0.92 | 0.87  | 0.64 | 0.20 | 0.72  | 0.67 |
> > | SPL        | 0.75 | 0.52 | 0.72  | 0.60 | 0.42 | 0.45  | 0.58 |
> > | DSR        | 0.20 | 0.11 | 0.46  | 0.76 | 0.72 | 0.75  | 0.50 |
> > | HIS (Ours) | **0.82** | **0.94** | **0.94**  | **0.83** | **0.99** | **0.75**  | **0.88** |
> >
> > [4]. Hoerl A E, Kennard R W. Ridge regression: applications to nonorthogonal problems. Technometrics, 1970.
> >
> > [5]. Chen T, Guestrin C. Xgboost: A scalable tree boosting system. Proceedings of the 22nd acm sigkdd international conference on knowledge discovery and data mining.
> >
> > [6]. d'Ascoli S, Renard A, Papadopoulos V, et al. Boolformer: Symbolic regression of logic functions with transformers. 2nd AI for Math Workshop@ICML 2025.
> >
> > [7]. Shirani Faradonbeh R, Monjezi M, Jahed Armaghani D. Genetic programing and non-linear multiple regression techniques to predict backbreak in blasting operation. Engineering with computers, 2016.
> >
> > [8]. Petersen B K, Larma M L, Mundhenk T N, et al. Deep symbolic regression: Recovering mathematical expressions from data via risk-seeking policy gradients. ICLR 2019.
> >
> > [9]. Sun F, Liu Y, Wang J X, et al. Symbolic Physics Learner: Discovering governing equations via Monte Carlo tree search. ICLR, 2024.
> >
> >
> > ### Question 1
> > > **Please respond to the concerns I have raised in the 'weaknesses' section. If the revision can adequately address most of the critical issues, I would consider raising the score.**
> >
> > We sincerely appreciate the reviewer for the thoughtful and constructive comments. The comments in the weaknesses section are **highly valuable for strengthening our work**, and we are carefully revising the manuscript to address each point in depth. **We truly appreciate your willingness to reconsider the score after the revision**. If there are any further questions or concerns, please don't hesitate to let us know, and we will continue actively responding to your comments and improving our submission.
> >
> > ### Question 2
> > > **In my personal opinion, the contribution of this paper lies more in its solid experimentation and reproducibility than in its theoretical advancement in ML. Considering EDA-focused conferences would be a more suitable venue for it.**
> >
> > We sincerely thank the reviewer for the thoughtful assessment and for highlighting the strengths of our experimental design and reproducibility. While our method indeed demonstrates strong applicability to EDA scenarios, we would like to respectfully clarify that **the core contribution of this work lies in its machine learning methodology rather than domain-specific engineering**.
> >
> > Specifically, our approach introduces:
> > - **A novel hierarchical symbolic tree representation**, which supports interpretable and efficient message aggregation and enables the model to capture rich structural information inherent in circuit graphs.
> > - **A Transformer-based symbolic discovery module**, which learns expressive symbolic scoring functions and generalizes across diverse optimization operators through its powerful sequence modeling capabilities.
> > - **A reinforcement learning–based training framework**, which adaptively guides the search toward high-quality symbolic expressions and improves both the robustness and generalization ability of the discovered scoring functions.
> >
> > These components contribute methodological insights that we believe are of broad relevance to the ML community. Moreover, **we respectfully think that the proposed learning paradigm and novel symbolic modeling approach constitute the primary conceptual advancement of this paper**. Overall, we appreciate the reviewer’s suggestion regarding alternative venues and hope this clarification helps illustrate the ML-centered contributions of our work.

---

> ### Author Response · Authors · 2025-11-26
> **Gentle Reminder: Rebuttal available and we sincerely look forward to your response**
>
> Dear Reviewer KPds,
>
> This is a gentle reminder that we have submitted our rebuttal and updated the paper accordingly. If you have a moment, we would greatly appreciate it if you could take a look and share any further comments during the discussion phase. We are very happy to clarify any points or provide additional information as needed.
>
> Thank you very much for your time and for engaging with our submission.
>
> Best regards,
>
> Authors of Submission 23006

---

### Official Review · Reviewer_DPdp · 2025-10-30

**Soundness:** 3
**Presentation:** 3
**Contribution:** 2
**Rating:** 6
**Confidence:** 3

**Summary:**

This paper proposes HIS (Hierarchical Circuit Symbolic Discovery Framework), a novel approach for efficient logic optimization in chip design. The key innovation lies in learning hierarchical symbolic functions that can capture circuit structural information while maintaining interpretability and computational efficiency. The framework uses structure-aware Transformers with reinforcement learning to generate symbolic trees that perform message passing similar to GNNs but with significantly lower inference cost. Experiments on EPFL and IWLS benchmarks demonstrate that HIS achieves runtime improvement and circuit size reduction when integrated with the Mfs2 heuristic, while being faster than GNN-based methods during inference.

**Strengths:**

(1) novel problem formulation: First GNN-free work to apply symbolic discovery to circuit graphs for logic optimization.
(2) Good originality: proposing a hierarchical symbolic tree structure that independently learns to mimic GNN-style message passing.
(3) Comprehensive Evaluation: The authors conducted a comprehensive empirical evaluation to validate their approach, including inference time, online/offline evaluation, QoR improvement etc.
(4) Clear and Intuitive Presentation: The paper presents a clear writing style and visual aids.

**Weaknesses:**

(1) Limited Analysis on Training Overhead: While the paper excellently highlights the inference speed-up, it lacks a analysis of the training time overhead, which can be substantial for reinforcement learning-based methods.
(2)  Inconsistent Performance Gains: The experimental results do not consistently demonstrate the superiority of HIS over the state-of-the-art CMO. While HIS shows an advantage in some specific settings (e.g., HIS-Mfs2 on 'Hyp' and 'DesPerf'), it underperforms CMO in QoR-focused scenarios (e.g., 2HIS-Mfs2), raising concerns about the stability and real-world benefit of the proposed method.
(3) The Claim of Interpretability is Not Substantiated by the Final Outputs: While HIS proposes a conceptually novel hierarchical framework, its final symbolic expressions seems are not demonstrably more interpretable than those from CMO.

**Questions:**

(1) What is the computational cost of training the hierarchical symbolic trees compared to GNN based symbolic discovery method such as CMO?
(2) Is the claimed interpretability advantage primarily from the hierarchical learning framework, or do the final, complex symbolic expressions offer more practical insights for domain experts than the simpler, non-hierarchical functions from methods like CMO?
(3) The paper focuses solely on the Mfs2 heuristic. How does the HIS framework generalize to other critical LO operators with different graph structures and optimization goals, such as Refactor?

---

> ### Author Response · Authors · 2025-11-22
> **Response to Reviewer DPdp---Part 1/5**
>
> Dear Reviewer DPdp,
>
> We greatly appreciate your careful reading and constructive feedback! We have provided further responses as follows. We sincerely hope that our responses could properly address all your concerns. If so, we would deeply appreciate it if you could raise your score. If not, please let us know your further concerns, and we will continue actively responding to your comments and improving our submission.
> ### Weakness 1 & Question 1
> > **What is the computational cost of training the hierarchical symbolic trees compared to GNN based symbolic discovery method such as CMO?**
>
> We sincerely appreciate the reviewer for the insightful question. We compare the computational cost of our HIS with the GNN-based method COG and the GNN-based symbolic discovery method CMO. All experiments are conducted on **four NVIDIA GeForce RTX 3090 GPUs**, and **the number of training steps is fixed at 2000** for GNN and our Transformer model. As shown in Table a, although our RL-based HIS requires slightly longer training time than the two baselines, **the overall resource cost remains modest, with a total training time of no more than 16 hours across both benchmarks**.
>
> **Table a:** We compare the training time of our HIS with other two GNN-based baselines on four NVIDIA GeForce RTX 3090 GPUs. Note that since CMO needs a GNN teacher, the reported training time for CMO includes both the GNN training phase and the subsequent symbolic discovery process.
> |            | EPFL              | IWLS               |
> |------------|-------------------|--------------------|
> | Method     | Training time (h) | Training time (h)  |
> | COG        | 8.18              | 9.95               |
> | CMO        | 10.38              | 12.35              |
> | HIS (Ours) | 13.76             | 15.37              |

---

> ### Author Response · Authors · 2025-11-22
> **Response to Reviewer DPdp---Part 2/5**
>
> ### Weakness 2
> > **Inconsistent Performance Gains: While HIS shows an advantage in some specific settings (e.g., HIS-Mfs2 on 'Hyp' and 'DesPerf'), it underperforms CMO in QoR-focused scenarios (e.g., 2HIS-Mfs2), raising concerns about the stability and real-world benefit of the proposed method.**
>
> We sincerely appreciate the reviewer for the valuable comments. As we did not provide the QoR results of CMO in our submission, **we respectfully assume that the reviewer referred to the online results of 2CMO-Mfs2 reported in [1]** for comparison with our 2HIS-Mfs2.
>
> However, we would like to clarify that **the comparison is not entirely fair due to differences in the evaluation strategies**. Specifically, the method in [1] uses an **in-domain generalization strategy**, where the test and training circuits are drawn from the same benchmark. In contrast, our HIS employs a **more challenging and real-world out-of-domain generalization strategy**, where the test circuits and training circuits come from different benchmarks.
>
> To ensure a fair comparison, we evaluate 2HIS-Mfs2 and 2CMO-Mfs2 under the same out-of-domain generalization strategy. As shown in Table b, **our 2HIS-Mfs2 outperforms 2CMO-Mfs2 in terms of optimization results**, including both circuit size and optimization time.
>
> **Table b:** We compare our HIS and CMO in terms of the online optimization results under the out-of-domain generalization strategy.
> |                  | **Hyp**           |                        |           |                         | **Multiplier**    |                       |           |                         |
> |------------------|---------------|------------------------|-----------|-------------------------|---------------|-----------------------|-----------|-------------------------|
> | Method           |  Lev          | Lev Impr (%) | Time (s)  | Time Impr (%) |  Nd | Nd Impr (%) | Time (s)  | Time Impr (%)|
> | The Default      | 8259.00       | NA                     | 265.93    | NA                      | 7799.00       | NA                    | 16.91     | NA                      |
> | 2CMO-Mfs2        | 5762.00       | 30.23                  | 204.38    | 23.15                   | 7661.00       | 1.77                  | 18.06     | -6.83                   |
> | 2HIS-Mfs2 (Ours) | 5762.00       | **30.23**                  | 147.62    | **44.49**                   | 7661.00       | **1.77**                  | 16.48     | **2.50**                    |
> |                  | **Square**        |                        |           |                         | **DesPerf**       |                       |           |                         |
> | Method           |  Nd | Nd Impr (%) | Time (s)  | Time Impr (%)| Nd | Nd Impr (%) | Time (s)  | Time Impr (%)|
> | The Default      | 5701.00       | NA                     | 21.48     | NA                      | 30853.00      | NA                    | 28.82     | NA                      |
> | 2CMO-Mfs2        | 5574.00              |2.23                        |16.64           |22.54                         | 30312.00      | 1.48                  | 18.05     | **37.37**                   |
> | 2HIS-Mfs2 (Ours) | 5553.00       | **2.60**                   | 16.21     | **24.54**                   | 30104.00      | **1.67**                  | 24.88     | **13.65**                   |
> |                  |**Ethernet**      |                        |           |                         | **Conmax**        |                       |           |                         |
> | Method           |  Nd | Nd Impr (%) | Time (s)  | Time Impr (%)| Nd | Nd Impr (%) | Time (s)  | Time Impr (%)|
> | The Default      | 13638.00      | NA                     | 27.00         | NA                   | 16509.00      | NA                    | 19.93        | NA                   |
> | 2CMO-Mfs2        |  13590.00             |  2.26                      |  20.51         |    24.05                     | 15792.00              |  **4.34**                    | 17.91          |      10.13                   |
> | 2HIS-Mfs2 (Ours) | 13511.00      | **4.34**                   | 14.84      | **45.06**                   | 15890.00      | ****1.79****                  |13.33       | **33.09**                   |
>
> [1]. Bai Y, Wang J, Chen L, et al. A Graph Enhanced Symbolic Discovery Framework For Efficient Logic Optimization. The Thirteenth International Conference on Learning Representations.

---

> ### Author Response · Authors · 2025-11-22
> **Response to Reviewer DPdp---Part 3/5**
>
> ### Weakness 3 and Question 2
> > **Is the claimed interpretability advantage primarily from the hierarchical learning framework, or do the final, complex symbolic expressions offer more practical insights for domain experts than the simpler, non-hierarchical functions from methods like CMO?**
>
> We sincerely appreciate the reviewer for the insightful questions. **Yes, our interpretability advantage primarily stems from our hierarchical learning framework and the inherent interpretability of symbolic representations.**
>
> First, compared with **black-box GNN** methods such as COG, our approach learns a **white-box symbolic function**. This function clearly shows how node features influence the final score, providing inherent interpretability that helps researchers better understand the model’s decision process for logic optimization.
>
> Second, compared with **node-based symbolic discovery methods such as CMO**, our approach learns a **graph-based symbolic function that provides richer, structure-aware interpretability**. As shown in Table c, three key observations emerge from the learned symbolic functions:
>
> - **Neighbor information aggregation improves expressive power:** Aggregating features from neighboring nodes significantly enhances the symbolic function’s ability to capture structural dependencies and model the underlying transformation patterns.
> - **Neighboring nodes exert equal influence on predictions for IWLS benchmark:** For IWLS benchmark, the use of the *mean* operator indicates that all neighboring nodes contribute uniformly, reflecting a symmetric and evenly weighted influence from the local subgraph.
> - **The root-node feature $H_{v_0}^2$ is particularly influential:** As shown in Table d, frequency analysis across all high-performance symbolic functions demonstrates that $H_{v_0}^2$ appears far more often than other features, highlighting its strong predictive importance.
> These findings demonstrate that our method can **explicitly characterize how nodes influence one another within the graph**, offering deeper interpretability than CMO.
>
> **Table c:** Comparison of the symbolic function discovered by our HIS and CMO.
> |      | EPFL                                                         |        | IWLS                                                         |        |
> | ---- | ------------------------------------------------------------ | ------ | ------------------------------------------------------------ | ------ |
> |      | Function                                                     | Recall | Function                                                     | Recall |
> | CMO  | $((((((H_{v_0}^2*H_{v_0}^3)+H_{v_0}^0)-\cos(H_{v_0}^4))-H_{v_0}^1)/H_{v_0}^0)-\sin((\exp(\cos(H_{v_0}^1))+H_{v_0}^3)))$ | 0.87   | $((H_{v_0}^0-\cos(H_{v_0}^3))-H_{v_0}^2)$                    | 0.7    |
> | HIS  | $((\min((H_{v_1}^4*H_{v_0}^3))+0.2+H_{v_0}^1)+H_{v_0}^2)$    | **0.9**    | $((H_{v_0}^2-(0.1+(H_{v_0}^2-mean((mean((H_{v_1}^4*H_{v_2}^1))+H_{v_0}^2)))))+H_{v_0}^1)$ | **0.86**   |
>
> **Table d:** We analyze the appearance frequency of node features across three layers within the high-performance symbolic functions generated by the best policy. The results highlight the critical importance of the second root-node feature $H_{v_0}^2$.
> |              | EPFL          | IWLS          |
> | ------------ | ------------- | ------------- |
> | Feature type | Frequency (%) | Frequency (%) |
> | $H_{v_0}^0$ (Fanin number)           | 6.29          | 4.97          |
> | $H_{v_0}^1$ (Fanout number)           | 14.58         | 15.80         |
> | $H_{v_0}^2$ (Level)           | **45.82**     | **41.39**     |
> | $H_{v_0}^3$ (Right level)           | 24.22         | 33.46         |
> | $H_{v_0}^4$ (Node id)            | 9.08          | 4.38          |

---

> ### Author Response · Authors · 2025-11-22
> **Response to Reviewer DPdp---Part 4/5**
>
> ### Question 3
> >**The paper focuses solely on the Mfs2 heuristic. How does the HIS framework generalize to other critical LO operators with different graph structures and optimization goals, such as Refactor?**
>
> We sincerely appreciate the reviewer’s insightful comments. Following your suggestions, we **extended our evaluation to mainstream pre-mapping heuristics**, which operate on **And-Inverter Graphs (AIGs) rather than K-LUTs** as in post-mapping heuristics such as Mfs2.
>
> Specifically, we conducted experiments on **one of the most challenging and time-consuming pre-mapping heuristics---Resub---as shown in Table e**. Moreover, since commonly used pre-mapping heuristics such as **Resub, Rewrite and Refactor follow the same overall paradigm** and differ primarily in their node-level transformation rules, our approach is naturally applicable to Refactor as well. **We will continue evaluating our method on other heuristics and include the complete results in the camera-ready version.**
>
> Specifically, we collect training data using the same pipeline as in the Mfs2 setting in the original submission: node features are extracted from the AIG, and node-level labels are obtained based on the effectiveness of Resub transformations. We then apply our HIS method to learn symbolic scoring functions. The results shown in **Table f** demonstrate that our method achieves the best performance on most of the circuits, **with an average prediction recall of 89%, surpassing other methods**. Moreover, the online results in Table g demonstrate that our method achieves **an average runtime reduction of 36.23% with only a marginal 0.40% degradation in circuit size** across the six test circuits. Therefore, we can conclude that our HIS **generalizes effectively to both pre-mapping heuristics (e.g., Resub) and post-mapping heuristics (e.g., Mfs2).**

---

> ### Author Response · Authors · 2025-11-22
> **Response to Reviewer DPdp---Part 5/5**
>
> **(continued part from "Response to Reviewer DPdp---Part 4/5")**
>
> **Table e:** We analyze the runtime of commonly used pre-mapping heuristics on six test circuits. Ratio denotes the ratio of average runtime to that of the Rewrite heuristic. Here $K$ represents the subgraph's relative size.
> | Avg Time Ratio to Rewrite |         |         |          |       |       |    |
> | ------------------------- | ------- | ------- | -------- | ----- | ----- | ----- |
> | **Heuristics**                | Rewrite | Balance | Refactor | Resub (K=12) | Resub (K=16) |Mfs2  |
> | **Time Ratio**                | 1       | 0.05    | 1.21     | **6.01** | **73.44** | 30.94 |
>
> **Table f:** The offline results show that HIS outperforms all baselines in terms of average generalization recall **for the Resub heuristic**.
> |            | Hyp      | Square   | Multiplier | Desperf  | Ethernet | Conmax   | Average  |
> | ---------- | -------- | -------- | ---------- | -------- | -------- | -------- | -------- |
> | **Method** | Recall   | Recall   | Recall     | Recall   | Recall   | Recall   | Recall   |
> | COG        | 0.87     | 0.89     | 0.87       | 0.72 | 0.92     | 0.76     | 0.84     |
> | Effisyn    | 0.67     | 0.63     | 0.20       | 0.46     | 0.82     | 0.02     | 0.47     |
> | Random     | 0.50     | 0.48     | 0.43       | 0.51     | 0.58     | 0.49     | 0.50     |
> | CMO        | 0.90     | 0.82     | 0.88       | 0.65     | 0.91     | 0.92     | 0.85     |
> | HIS（Ours）        | 0.92 | 0.93 | 0.93   | 0.68     | 0.93 | 0.96 | **0.89** |
>
> **Table g:** We compare the Default Resub heuristic with our HIS-Resub heuristic with the hyperparameter $k$ set $50$% on six challenging circuits.
> |             | **Hyp**      |                       |          |                         | **Multiplier** |                       |          |                         |
> | ----------- | ------------ | --------------------- | -------- | ----------------------- | -------------- | --------------------- | -------- | ----------------------- |
> | Method      | Nd           | Nd Impr (%)$\uparrow$ | Time (s) | Time Impr (%)$\uparrow$ | Nd             | Nd Impr (%)$\uparrow$ | Time (s) | Time Impr (%)$\uparrow$ |
> | The Default | 204533.00    | NA                    | 161.82   | NA                      | 24436.00       | NA                    | 33.47    | NA                      |
> | HIS-Resub   | 205257.00    | **-0.34**             | 67.16    | **58.50**               | 24453.00       | **-0.07**                 | 25.07    | **25.11**               |
> |             | **Square**   |                       |          |                         | **DesPerf**    |                       |          |                         |
> | Method      | Nd           | Nd Impr (%)$\uparrow$ | Time (s) | Time Impr (%)$\uparrow$ | Nd             | Nd Impr (%)$\uparrow$ | Time (s) | Time Impr (%)$\uparrow$ |
> | The Default | 15845.00     | NA                    | 14.31    | NA                      | 67193.00       | NA                    | 302.69   | NA                      |
> | HIS-Resub   | 15983.00     | **-0.83**             | 7.91     | **44.73**               | 67892.00       | **-1.01**             | 242.93   | **19.74**               |
> |             | **Ethernet** |                       |          |                         | **Conmax**     |                       |          |                         |
> | Method      | Nd           | Nd Impr (%)$\uparrow$ | Time (s) | Time Impr (%)$\uparrow$ | Nd             | Nd Impr (%)$\uparrow$ | Time (s) | Time Impr (%)$\uparrow$ |
> | The Default | 43345.00     | NA                    | 109.94   | NA                      | 39126.00       | NA                    | 156.54   | NA                      |
> | HIS-Resub   | 43345.00     | **0.00**              | 82.65    | **45.06**               | 39195.00       | **-0.17**             | 118.59   | **24.24**               |

---

> ### Author Response · Authors · 2025-11-26
> **Gentle Reminder: Rebuttal available and we sincerely look forward to your response**
>
> Dear Reviewer DPdp,
>
> This is a gentle reminder that we have submitted our rebuttal and updated the paper accordingly. If you have a moment, we would greatly appreciate it if you could take a look and share any further comments during the discussion phase. We are very happy to clarify any points or provide additional information as needed.
>
> Thank you very much for your time and for engaging with our submission.
>
> Best regards,
>
> Authors of Submission 23006

---

### Official Review · Reviewer_63LY · 2025-10-30

**Soundness:** 3
**Presentation:** 2
**Contribution:** 3
**Rating:** 6
**Confidence:** 4

**Summary:**

This paper proposes a novel Hierarchical Circuit Symbolic Discovery Framework (HIS) for efficient logic optimization. HIS aims to address the high inference cost and limited interpretability of existing machine-learning cost-prediction methods, such as GNNs. The framework leverages reinforcement learning to train a structure-aware Transformer model that automatically discovers a lightweight and interpretable symbolic function. This function, represented as a hierarchical tree, mimics the message-passing of GNNs to identify ineffective node-level transformations for pruning. Extensive experiments on two public benchmarks demonstrate that HIS outperforms state-of-the-art methods in prediction accuracy and optimization efficiency. When integrated with the Mfs2 heuristic, HIS significantly reduces runtime while maintaining or even improving optimization quality.

**Strengths:**

1. **Well-identified the practical problems of GNNs:** The paper correctly pinpoints the critical bottlenecks— **high inference cost and lack of interpretability and** —that prevent state-of-the-art GNNs from being widely deployed in real-world EDA tools. This shows a clear understanding of industrial needs beyond just academic metrics.
2. **Innovative methodology combining generation and RL:** The core idea of using a **generative model (Transformer) coupled with Reinforcement Learning** to discover an interpretable symbolic tree is highly innovative. It reframes the problem from a black-box prediction task to a structured search for a human-readable function, effectively bridging the gap between performance and interpretability.
3. **Solid integration with industry tools and demonstrated gains:** The research is grounded in practicality. By seamlessly integrating with a standard EDA framework (ABC) and a critical, time-consuming heuristic (Mfs2), and demonstrating **significant, measurable improvements** (27.22% speedup, 6.95% size reduction), the work is far more **substantial and deployable** compared to many theoretical GNN studies that lack a clear path to real-world application.

**Weaknesses:**

1. **Misleading figures and poor readability:** The writing and diagrams are currently a significant weakness and hinder understanding.
    - **Figures:** Figure 1 mentions "GNNs" in a way that is confusing and seems disconnected from the HIS methodology. In Figure 2, the flow of "Generation Process" and "Model Training" are not expressive and hard to understand.
    - **Writing:** The methodology section requires significant polishing. It needs to guide the reader through the complex pipeline in a simpler, more logical, and step-by-step manner. The current presentation is dense and difficult to follow.
2. **Omission of training cost:** While the paper justifiably highlights the model's superior **inference efficiency**, it omits any discussion of the **training time and computational resources** required. For a comprehensive cost-benefit analysis, especially for industrial adoption, the substantial computational overhead of training the Transformer with RL must be considered and disclosed.
3. **Non-actionable interpretability:** The claimed "interpretability" is superficial in its current form. While the final symbolic tree is indeed human-readable, the pipeline does **not leverage this interpretability to adjust or improve the model**. It is a one-way, post-hoc explanation. There is no feedback loop where a human expert can correct or guide the symbolic discovery process based on the generated formulas.
4. **Lack of qualitative analysis:** The authors missed a key opportunity to use interpretability for a compelling **qualitative analysis**. The paper lacks a case study or an in-depth discussion that answers questions like: "What did we learn from the discovered formulas?" or "Can we see a specific example where the symbolic tree correctly identified a complex structural pattern that a human would have missed?" Including such an analysis would have powerfully demonstrated the unique advantage of their method over pure black-box approaches.

**Questions:**

1. **Scalability of the Symbolic Tree:** The paper demonstrates the effectiveness of relatively small symbolic trees for integration into existing tools. Could the authors comment on the scalability of their approach? What would be the computational cost and potential performance implications of training a model to discover significantly larger and more complex symbolic expressions?
2. **Scaling Laws and Potential for a Foundational Model:** A key question for the long-term impact of this methodology is its adherence to scaling laws. Do the model's performance and generalization capability improve predictably with increased model capacity and training data volume? If empirical evidence supported such scaling laws, it would justify the development of a large-scale, pre-trained symbolic model for EDA. Can the authors provide any insight or preliminary results on this front?
3. **Sensitivity to the Symbolic Library:** The choice of the symbolic library is a strong prior in the discovery process. The results appear sensitive to this choice, as more complex operators (e.g., `log`, `exp`) are available yet absent from the discovered functions in Figure 5. A dedicated ablation study on the composition of the symbolic library is lacking. How do the results change if key operator types (e.g., aggregation functions) are removed? Such a study is crucial for understanding the true expressiveness of the discovered functions and the biases introduced by the library.

---

> ### Author Response · Authors · 2025-11-22
> **Response to Reviewer 63LY---Part 1/4**
>
> Dear Reviewer 63LY,
>
> We greatly appreciate your careful reading and constructive feedback! We have provided further responses as follows. We sincerely hope that our responses could properly address all your concerns. If so, we would deeply appreciate it if you could raise your score. If not, please let us know your further concerns, and we will continue actively responding to your comments and improving our submission.
> ### Weakness 1
> > **Misleading figures and poor readability:
> (1). Figures: Figure 1 mentions "GNNs" in a way that is confusing and seems disconnected from the HIS methodology. In Figure 2, the flow of "Generation Process" and "Model Training" are not expressive and hard to understand.
> (2). Writing: The methodology section requires significant polishing. It needs to guide the reader through the complex pipeline in a simpler, more logical, and step-by-step manner. The current presentation is dense and difficult to follow.**
>
> We sincerely appreciate the reviewer’s valuable comments. In response, **we have carefully revised Figures 1 and 2 in the updated manuscript** to remove potential sources of confusion and improve their clarity. Regarding the writing issues, **we will thoroughly refine the methodology section**, restructuring it into a clearer and more logically organized step-by-step presentation to further enhance readability.
>
> Once again, we would like to thank the reviewer for these constructive and helpful suggestions.
>
> ### Weakness 2 and Question 1
> >**Scalability of the Symbolic Tree: It omits any discussion of the training time and computational resources required. Moreover, What would be the computational cost and potential performance implications of training a model to discover significantly larger and more complex symbolic expressions?**
>
> We sincerely thank the reviewer for the insightful question. We compare the computational cost of our HIS with the GNN-based method COG and the GNN-based symbolic discovery method CMO. All experiments are conducted on **four NVIDIA GeForce RTX 3090 GPUs**, and **the number of training steps is fixed at 2000** for GNN and our Transformer model. As shown in Table a, although our **RL-based** method requires slightly longer training time than the two baselines, **the overall resource cost remains modest, with a total training time of no more than 16 hours across both benchmarks**.
>
> Moreover, to examine how computational cost scales with the complexity of the target function, we varied the function length constraint from 50 to 150. As shown in Table b, obtaining a final symbolic function **with length/complexity of 86**---which is substantially higher than the complexity of the functions ultimately discovered in our experiments with length/complexity of 10---requires **around 20 hours of computation on four NVIDIA 3090 GPUs**. However, **despite this substantial increase in expression complexity, the performance improves only marginally**, indicating that lightweight hierarchical symbolic expressions are already sufficient for the logic synthesis proplem.
>
> **Table a:** We compare the training time of our HIS with other two GNN-based baselines on four NVIDIA GeForce RTX 3090 GPUs. Note that since CMO needs a GNN teacher, the reported training time for CMO includes both the GNN training phase and the subsequent symbolic discovery process.
> |            | EPFL              | IWLS               |
> |------------|-------------------|--------------------|
> | Method     | Training time (h) | Training time (h)  |
> | COG        | 8.18              | 9.95               |
> | CMO        | 10.38              | 12.35              |
> | HIS (Ours) | 13.76             | 15.37              |
>
> **Table b:** Training time required to discover a highly complex symbolic function. Here, *complexity* refers to the number of symbol tokens in the best generated function.
> |                       | EPFL       |                   | |IWLS       |                   ||
> | --------------------- | ---------- | ----------------- | ---------- | ----------------- | ---------- | ----------------- |
> | **Constraint Length** | Complexity | Training time (h) | Recall| Complexity | Training time (h) |Recall|
> | Length=50             | 12         | 13.76          |  0.90  | 10         | 15.37             | 0.86
> | Length=200            | 50         | 17.63         | 0.90     | 86         | 19.62             | 0.87

---

> ### Author Response · Authors · 2025-11-22
> **Response to Reviewer 63LY---Part 2/4**
>
> ### Weakness 4
> >**The paper lacks a case study or an in-depth discussion that answers questions like: "What did we learn from the discovered formulas?" or "Can we see a specific example where the symbolic tree correctly identified a complex structural pattern that a human would have missed?"**
>
> We sincerely thank the reviewer for the insightful and valuable comments. Following your suggestions, we would like to **clarify the interpretability advantage of our method** and **summarize the key insights obtained from the discovered symbolic formulas**.
>
> **The interpretability advantage:** Our interpretability advantage primarily stems from our hierarchical learning framework and the inherent interpretability of symbolic representations. First, compared with **black-box GNN** methods such as COG, our approach learns a **white-box symbolic function**. This function clearly shows how node features influence the final score, providing inherent interpretability that helps researchers better understand the model’s decision process for logic optimization. Second, compared with **node-based symbolic discovery methods such as CMO**, our approach learns a **graph-based symbolic function that provides richer, structure-aware interpretability**.
>
> **Key insights from the interpretable symbolic functions:** As shown in Table c, the learned symbolic functions reveal the following three core insights:
>
> - **Neighbor information aggregation improves expressive power:** Aggregating features from neighboring nodes significantly enhances the symbolic function’s ability to capture structural dependencies and model the underlying transformation patterns.
> - **Neighboring nodes exert equal influence on predictions for IWLS benchmark:** For IWLS benchmark, the use of the *mean* operator indicates that all neighboring nodes contribute uniformly, reflecting a symmetric and evenly weighted influence from the local subgraph.
> - **The root-node feature $H_{v_0}^2$ is particularly influential:** As shown in Table d, frequency analysis across all high-performance symbolic functions demonstrates that $H_{v_0}^2$ appears far more often than other features, highlighting its strong predictive importance.
>
>
> These findings demonstrate that our method can **explicitly characterize how nodes influence one another within the graph**, offering deeper interpretability than existing logic optimization approaches.
>
> **Table c:** Comparison of the graph symbolic function discovered by our HIS with the node-based symbolic function generated by CMO.
> |      | EPFL                                                         |        | IWLS                                                         |        |
> | ---- | ------------------------------------------------------------ | ------ | ------------------------------------------------------------ | ------ |
> |      | Function                                                     | Recall | Function                                                     | Recall |
> | CMO  | $((((((H_{v_0}^2*H_{v_0}^3)+H_{v_0}^0)-\cos(H_{v_0}^4))-H_{v_0}^1)/H_{v_0}^0)-\sin((\exp(\cos(H_{v_0}^1))+H_{v_0}^3)))$ | 0.87   | $((H_{v_0}^0-\cos(H_{v_0}^3))-H_{v_0}^2)$                    | 0.7    |
> | HIS  | $((\min((H_{v_1}^4*H_{v_0}^3))+0.2+H_{v_0}^1)+H_{v_0}^2)$    | **0.9**    | $((H_{v_0}^2-(0.1+(H_{v_0}^2-mean((mean((H_{v_1}^4*H_{v_2}^1))+H_{v_0}^2)))))+H_{v_0}^1)$ | **0.86**   |
>
> **Table d:** We analyze the appearance frequency of node features across three layers within the high-performance symbolic functions generated by the best policy. The results highlight the critical importance of the second root-node feature $H_{v_0}^2$.
> |              | EPFL          | IWLS          |
> | ------------ | ------------- | ------------- |
> | Feature type | Frequency (%) | Frequency (%) |
> | $H_{v_0}^0$ (Fanin number)           | 6.29          | 4.97          |
> | $H_{v_0}^1$ (Fanout number)           | 14.58         | 15.80         |
> | $H_{v_0}^2$ (Level)           | **45.82**     | **41.39**     |
> | $H_{v_0}^3$ (Right level)           | 24.22         | 33.46         |
> | $H_{v_0}^4$ (Node id)            | 9.08          | 4.38          |

---

> ### Author Response · Authors · 2025-11-22
> **Response to Reviewer 63LY---Part 3/4**
>
> ### Weakness 3
> >**The pipeline does not leverage the interpretability of the final formulas to adjust or improve the model.**
>
> We sincerely appreciate the reviewer for the insightful and valuable comments. Building on the interpretability findings discussed in *Weakness 4*, we further **investigate how to enhance the contribution of the second root-node feature $H_{v_0}^2$**. Specifically, we **introduce an additional encoding vector to the input token corresponding to $H_{v_0}^2$**, thereby emphasizing its importance during current token generation. As shown in Table e, this modification increases the occurrence of $H_{v_0}^2$ in the learned symbolic functions and leads to a light average performance improvement.
>
> **Table e:** We leverage the interpretability insight regarding the critical role of $H_{v_0}^2$ to guide our model design. The results demonstrate the effectiveness of our adjustment.
> |                           | Hyp    | Multiplier | Square | DesPerf | Ethernet | Conmax | Average  |
> | ------------------------- | ------ | ---------- | ------ | ------- | -------- | ------ | -------- |
> | Method                    | Recall | Recall     | Recall | Recall  | Recall   | Recall | Recall   |
> | HIS                       | 0.82   | 0.94       | 0.94   | 0.83    | 0.99     | 0.75   | **0.88** |
> | HIS with feature encoding | 0.82   | 0.94       | 0.96   | 0.83    | 0.92     | 0.94   | **0.90** |
>
> ### Question 2
> > **Scaling Laws and Potential for a Foundational Model: Do the model's performance and generalization capability improve predictably with increased model capacity and training data volume?**
>
> We sincerely appreciate the reviewer’s insightful and valuable questions. In response, we conducted two groups of experiments to systematically investigate the interplay between model capacity, training data volume, and prediction performance.
>
> **First**, we increased the size of the training dataset from $10^2$ to $10^4$. As shown in Table f, the average prediction performance on two benchmarks **exhibit a clear and stable scaling trend with respect to data volume**. **Second**, while fixing the data size at $3*10^4$, we scaled the model capacity by varying the number of Transformer encoder layers to 4, 6, and 8. As reported in Table g, **increasing the model depth does not yield further performance improvement**.
>
> Overall, our results demonstrate that scaling model capacity dosen't yield benefits, while increasing the training data volume consistently enhances performance, **highlighting data—not model size—as the primary determinant of effectiveness for our symbolic discovery task.**
>
> **Table f:** We analyze the effect of the training data volume on final prediction performance.
> |                     | EPFL     |                   | IWLS   |                   |
> | ------------------- | -------- | ----------------- | ------ | ----------------- |
> | Traning data number | Recall   | Training Time (h) | Recall | Training Time (h) |
> | 10^2                | 0.76     | 10.80             | 0.80   | 12.49             |
> | 10^3                | 0.78     | 11.05             | 0.82   | 13.98             |
> | 10^4                | 0.89     | 12.56             | 0.84   | 14.62             |
> | 3*10^4              | **0.90** | 13.76             | **0.86**   | 15.37             |
>
> **Table g:** We analyze the effect of the model size (i.e., Transformer encoder layers) on final prediction performance.
> |                   | EPFL     |                   | IWLS     |                   |
> | ----------------- | -------- | ----------------- | -------- | ----------------- |
> | Transformer Layer | Recall   | Training Time (h) | Recall   | Training Time (h) |
> | 4                 | **0.90** | 13.76             | **0.86** | 15.37             |
> | 6                 | 0.90     | 16.00             | 0.85     | 16.60             |
> | 8                 | 0.89     | 18.91             | 0.85     | 19.22             |

---

> > ### Author Response · Authors · 2025-11-22
> > **Response to Reviewer 63LY---Part 4/4**
> >
> > ### Question 3
> > > **Sensitivity to the Symbolic Library: How do the results change if key operator types (e.g., aggregation functions) are removed?**
> >
> > We sincerely appreciate the reviewer’s insightful and valuable questions. In response, we conducted a set of ablation studies to **assess the role of key operator types in our symbolic library**.
> >
> > In general, the operators can be categorized into two groups: basic operators {$+, -, \times, \div$} and complex operators {$\exp, \log, \min, \max, \mathrm{sum}, \mathrm{mean}$}. To quantify the contribution of the complex operators, we designed three variants: **HIS, HIS w/o aggregation, and HIS w/o $\log$ and $\exp$**. The results in Table h reveal two primary findings: (1). ***HIS w/o aggregation*** exhibits a substantial degradation in prediction performance, indicating that aggregation functions are essential for approximating the message-passing behavior in GNNs. This further confirms that **aggregation functions are the major source of the effectiveness of our hierarchical symbolic functions.** (2). ***HIS w/o $\log$ and $\exp$*** achieves performance comparable to the full HIS model, suggesting that **once aggregation functions are available, additional complex mathematical operators such as log and exp offer limited marginal benefit**.
> >
> >
> > **Table h:** The ablation study to assess the role of key operators in our symbolic library.
> > |                           | Hyp      | Multiplier | Square   | DesPerf  | Ethernet | Conmax   |
> > | ------------------------- | -------- | ---------- | -------- | -------- | -------- | -------- |
> > | Method                    | Recall    | Recall      | Recall    | Recall    | Recall    | Recall    |
> > | HIS                       | **0.82** | 0.94       | **0.94** | **0.83** | **0.99** | **0.75** |
> > | HIS w/o aggregation | 0.79     | 0.52       | 0.78     | 0.81     | 0.56     | 0.68     |
> > | HIS w/o $\exp$ and $\log$           | **0.82** | **0.95**   | 0.93     | 0.8      | 0.88     | **0.75** |

---

> ### Author Response · Authors · 2025-11-26
> **Gentle Reminder: Rebuttal available and we sincerely look forward to your response**
>
> Dear Reviewer 63LY,
>
> This is a gentle reminder that we have submitted our rebuttal and updated the paper accordingly. If you have a moment, we would greatly appreciate it if you could take a look and share any further comments during the discussion phase. We are very happy to clarify any points or provide additional information as needed.
>
> Thank you very much for your time and for engaging with our submission.
>
> Best regards,
>
> Authors of Submission 23006

---

### Official Review · Reviewer_L1RN · 2025-11-01

**Soundness:** 3
**Presentation:** 3
**Contribution:** 3
**Rating:** 6
**Confidence:** 2

**Summary:**

Existing logic optimization tools suffer from high inference costs and limited interpretability. The authors developed HIS, an effective logic optimization model that leverages hierarchical symbolic function representation and a corresponding group reward mechanism.

**Strengths:**

* The paper is well written, easy to follow
* Authors performed various experiments including ablation studies.
* The problem the authors aimed to address is practical and addresses a real demand in the field.

**Weaknesses:**

* I haven’t read many AI papers on logic optimization, so I’m not very familiar with the field, but the number of baselines seems rather limited. It would be beneficial to include more general AI models as baselines, even if they were not originally designed for logic optimization, as long as they can be applied to this problem.
* It appears that the experiments were conducted only once and the performance was reported based on that single run. To ensure that the model’s performance is not dependent on a specific random seed but is statistically meaningful, it is necessary to repeat the experiments multiple times and report the mean and standard deviation of the performance.
* From Table 3, it can be seen that for the Hyp, Multiplier, Square, and Conmax circuits, the performance difference with or without group optimization is not very large, whereas for DesPerf and Ethernet, the difference is significant. This suggests that while certain components of the model are highly beneficial for some circuits, they contribute little to others. It would be helpful if the authors could provide an explanation of whether this variation is related to the characteristics of the circuits.

**Questions:**

See the 'weakness' part

---

> ### Author Response · Authors · 2025-11-22
> **Response to Reviewer L1RN---Part 1/2**
>
> Dear Reviewer L1RN,
>
> We greatly appreciate your careful reading and constructive feedback! We have provided further responses as follows. We sincerely hope that our responses could properly address all your concerns. If so, we would deeply appreciate it if you could raise your score (or confidence). If not, please let us know your further concerns, and we will continue actively responding to your comments and improving our submission.
> ### Weakness 1
> >**The number of baselines seems rather limited. It would be beneficial to include more general AI models as baselines, even if they were not originally designed for logic optimization, as long as they can be applied to this problem.**
>
> We sincerely appreciate the reviewer for the thoughtful and constructive comments. Following your suggestion, we have incorporated **an extensive comparison with some recent methods**. Specifically, the baselines include **two traditional lightweight machine learning models**---RidgeLR[4], XGBoost[5]---and four powerful symbolic regression methods---Boolformer[6], GPLearn[7], DSR[8], SPL[9]. The offline results in Table a demonstrate that **our method significantly outperforms all baselines in terms of the prediction recall**, indicating the effectiveness of our method.
>
> **Table a:** We compare our method with more baselines in terms of the offline prediction recall.
> |  | Hyp      | Multiplier | Square   | DesPerf | Ethernet | Conmax | Average |
> | ------------------ | -------- | ---------- | -------- | -------- | -------- | --------- | --------- |
> | Method             | Recall    | Recall      | Recall    | Recall    | Recall    | Recall     | Recall     |
> | RidgeLR    | 0.74 | 0.62 | 0.88  | 0.79 | 0.33 | 0.54  | 0.65 |
> | XGBoost    | 0.71 | 0.86 | 0.46  | 0.79 | 0.33 | 0.68  | 0.64 |
> | Boolformer | 0.35 | 0.65 | 0.80  | 0.65 | 0.69 | 0.62  | 0.63 |
> | GPLearn    | 0.65 | 0.92 | 0.87  | 0.64 | 0.20 | 0.72  | 0.67 |
> | SPL        | 0.75 | 0.52 | 0.72  | 0.60 | 0.42 | 0.45  | 0.58 |
> | DSR        | 0.20 | 0.11 | 0.46  | 0.76 | 0.72 | 0.75  | 0.50 |
> | HIS (Ours) | **0.82** | **0.94** | **0.94**  | **0.83** | **0.99** | **0.75**  | **0.88** |
>
> [4]. Hoerl A E, Kennard R W. Ridge regression: applications to nonorthogonal problems. Technometrics, 1970.
>
> [5]. Chen T, Guestrin C. Xgboost: A scalable tree boosting system. Proceedings of the 22nd acm sigkdd international conference on knowledge discovery and data mining.
>
> [6]. d'Ascoli S, Renard A, Papadopoulos V, et al. Boolformer: Symbolic regression of logic functions with transformers. 2nd AI for Math Workshop@ICML 2025.
>
> [7]. Shirani Faradonbeh R, Monjezi M, Jahed Armaghani D. Genetic programing and non-linear multiple regression techniques to predict backbreak in blasting operation. Engineering with computers, 2016.
>
> [8]. Petersen B K, Larma M L, Mundhenk T N, et al. Deep symbolic regression: Recovering mathematical expressions from data via risk-seeking policy gradients. ICLR 2019.
>
> [9]. Sun F, Liu Y, Wang J X, et al. Symbolic Physics Learner: Discovering governing equations via Monte Carlo tree search. ICLR, 2024.
>
>
> ### Weakness 2
> >**To ensure that the model’s performance is not dependent on a specific random seed but is statistically meaningful, it is necessary to repeat the experiments multiple times and report the mean and standard deviation of the performance.**
>
> We sincerely thank the reviewer for the valuable comments. Following your suggestion, we repeated the main offline experiment **three times using different random seeds**. The results in Table b report the **mean and standard deviation across these runs, demonstrating the robustness and stability of our method**.
>
> **Table b:** The offline results of our HIS over three runs with different random seeds. The small standard deviation highlights the robustness of our method.
> |               | Hyp      | Multiplier | Square   | DesPerf  | Ethernet | Conmax   |
> | ------------- | -------- | ---------- | -------- | -------- | -------- | -------- |
> | Method        | Recall   | Recall     | Recall   | Recall   | Recall   | Recall   |
> | HIS(seed=0) | 0.82     | 0.94       | 0.94     | 0.83     | 0.99     | 0.75     |
> | HIS(seed=1) | 0.82     | 0.91       | 0.95     | 0.80     | 0.98     | 0.80     |
> | HIS(seed=2) | 0.84     | 0.91       | 0.95     | 0.80     | 0.94     | 0.80     |
> | **Average**   | **0.83** | **0.92**   | **0.95** | **0.81** | **0.97** | **0.78** |
> | **Std**       | **0.01** | **0.01**   | **0.00** | **0.01** | **0.02** | **0.02** |

---

> ### Author Response · Authors · 2025-11-22
> **Response to Reviewer L1RN---Part 2/2**
>
> ### Weakness 3
> >**From Table 3, it can be seen that for some circuits, the performance difference with or without group optimization is not very large, whereas for DesPerf and Ethernet, the difference is significant. It would be helpful if the authors could provide an explanation of whether this variation is related to the characteristics of the circuits.**
>
> We sincerely appreciate the reviewer for the insightful and valuable suggestions. Our further analysis indicates that the varying benefit of group optimization is closely related to the ***layer density*** of a circuit, defined as **the ratio between the total number of nodes and the number of logic levels**.
>
> As shown in Tale c, circuits such as **DesPerf** and **Ethernet** exhibit **high layer density**, meaning that each logic layer contains a large number of structurally similar nodes. In such cases, **many nodes share highly similar local structures, making their individual feature representations difficult to distinguish** and leading to ambiguity in score prediction. **The group optimization module alleviates this issue by introducing relative group-wise information**, which effectively separates these structurally similar nodes and enables more accurate scoring and optimization.
>
>
> In contrast, other circuits have **low layer density**, where each logic level contains fewer nodes and exhibits less structural redundancy. Consequently, node-level features are already sufficiently discriminative, and the additional relational cues introduced by group optimization bring only marginal benefits.
>
> **Table c:** The Layer density ($=\frac{Node\space number}{level}$) of the test circuits.
> |                   | DesPerf    | Ethernet   | Hyp     | Square   | Multiplier |Conmax |
> | ----------------- | ---------- | ---------- | ------- | -------- | ---------- | ---------- |
> | **Layer density** | **3516.5** | **1265.0** | 8.6 | 73.9 | 98.8   | 116.8

---

> ### Author Response · Authors · 2025-11-26
> **Gentle Reminder: Rebuttal available and we sincerely look forward to your response**
>
> Dear Reviewer L1RN,
>
> This is a gentle reminder that we have submitted our rebuttal and updated the paper accordingly. If you have a moment, we would greatly appreciate it if you could take a look and share any further comments during the discussion phase. We are very happy to clarify any points or provide additional information as needed.
>
> Thank you very much for your time and for engaging with our submission.
>
> Best regards,
>
> Authors of Submission 23006

---

### Meta-Review · Area_Chair_AHiu · 2026-01-07

**Summary:**

### Summary
This paper proposes HIS, a hierarchical circuit symbolic discovery framework for efficient logic optimization. HIS uses a structure-aware Transformer trained with reinforcement learning to discover lightweight, interpretable hierarchical symbolic functions that mimic GNN-style message passing while reducing inference cost. Experiments on EPFL and IWLS show strong offline prediction recall and, when integrated into ABC heuristics (notably Mfs2, and additionally Resub in the rebuttal), HIS achieves substantial runtime reduction with comparable or improved optimization quality.

### Strengths
- Practical motivation and impact: targets real deployment bottlenecks of GNN-based LO tools (inference cost, interpretability) and demonstrates measurable runtime gains in standard toolchains.
- Methodological novelty: reframes LO scoring from black-box prediction to structure-aware symbolic function discovery with hierarchical aggregation.
- Strong empirical coverage: ablations and multiple circuits/benchmarks; rebuttal adds broader baseline comparisons and multi-seed statistics.
- Credible generality evidence: rebuttal extends to pre-mapping heuristic (Resub) and provides operator-library ablations and scaling observations.

### Weaknesses
- Presentation/readability issues in the original submission (figures and dense methodology) could hinder accessibility, though authors commit to revisions.
- Training overhead and cost-benefit narrative was initially incomplete; rebuttal provides GPU/time tables but camera-ready should clearly summarize settings and costs.
- Interpretability remains partly under-exploited: despite added qualitative/insight analyses and a small feedback-loop tweak, more concrete case studies would strengthen the “actionable interpretability” claim.
- Some gains vary by circuit; rebuttal offers a layer-density explanation, but it would help to tie this to practical guidance on when group optimization is most beneficial.

HIS addresses a practically important logic-optimization bottleneck by learning lightweight, interpretable symbolic scoring functions that substantially reduce inference/runtime while maintaining optimization quality on established benchmarks. Across reviewers, the core method is viewed as novel and deployable, and the rebuttal strengthens the evaluation by adding more baselines, repeated-seed stability, training-cost disclosure, operator ablations, and an additional heuristic (Resub). While clarity and deeper interpretability analysis can be further improved in the final version, the technical contribution and demonstrated practical gains support acceptance.

**Reviewer Concerns:**

- L1RN
  - addressed: added broader baselines (traditional ML + symbolic regression); repeated runs with mean/std; explained circuit-dependent gains via layer density and redundancy.
  - still outstanding: none major; baseline breadth and statistical robustness concerns are substantially resolved.

- 63LY
  - addressed: added training-time disclosure and scaling-with-complexity experiment; committed figure/methodology rewriting; added qualitative interpretability insights + feature-frequency analysis; added operator-library ablation; explored data/model scaling; added a small interpretability-driven feedback tweak.
  - still outstanding: interpretability is improved but still limited in terms of concrete, end-to-end case studies demonstrating “what engineers should do differently”; presentation fixes need to be verified in the revised manuscript.

- DPdp
  - addressed: provided training-time comparisons vs COG/CMO; clarified fairness of QoR comparisons and added out-of-domain evaluation; strengthened interpretability discussion and comparisons; added Resub experiments and argued applicability to other pre-mapping heuristics.
  - still outstanding: stability/generalization beyond Mfs2/Resub is promising but still limited; camera-ready should more explicitly define evaluation protocols and when to expect QoR vs speed trade-offs.

- KPds
  - addressed (from shown rebuttal content): added a formal approximation theorem for GNN message passing; provided rationale/analysis for complex operators (log/exp) and later operator ablations (elsewhere in rebuttal thread).
  - still outstanding: broader validation across more heuristics and real industrial data remains a longer-term request; depth-L sensitivity and circuit-type (combinational vs sequential) analysis would still strengthen the paper if space permits.

**Reviewer Scores:**

- L1RN: would increase (key concerns on baselines, stability, and circuit variation were directly addressed; reviewer’s initial score was already above threshold but confidence low)
- 63LY: no change (major concerns were addressed in rebuttal, but reviewer’s score was already 6)
- DPdp: no change (rebuttal addresses fairness/training/interpretability/generalization)
- KPds: no change

---

### Decision · Program_Chairs · 2026-01-26

Accept (Poster)